# ReSpace: Text-Driven 3D Indoor Scene Synthesis and Editing with Preference Alignment

## Abstract

Scene synthesis and editing has emerged as a promising direction in computer graphics. Current trained approaches for 3D indoor scenes either oversimplify object semantics through one-hot class encodings (e.g., 'chair' or 'table'), require masked diffusion for editing, ignore room boundaries, or rely on floor plan renderings that fail to capture complex layouts. LLM-based methods enable richer semantics via natural language (e.g., 'modern studio with light wood furniture'), but lack editing functionality, are limited to rectangular layouts, or rely on weak spatial reasoning from implicit world models. We introduce ReSpace, a generative framework for text-driven 3D indoor scene synthesis and editing using autoregressive language models. Our approach features a compact structured scene representation with explicit room boundaries that enables asset-agnostic deployment and frames scene editing as a next-token prediction task. We leverage a dual-stage training approach combining supervised fine-tuning and preference alignment, enabling a specially trained language model for object addition that accounts for user instructions, spatial geometry, object semantics, and scene-level composition. For scene editing, we employ a zero-shot LLM to handle object removal and prompts for addition. We further introduce a voxelization-based evaluation capturing fine-grained geometry beyond 3D bounding boxes. Experimental results surpass state-of-the-art on addition and achieve superior human-perceived quality on full scene synthesis.

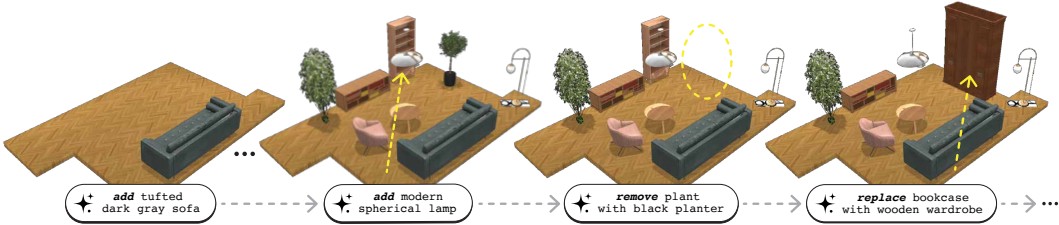

Figure 1: We introduce a novel text-driven framework for 3D indoor scene synthesis, completion, and editing—supporting object addition, removal, and swapping via natural language prompts.

## 1 Introduction

Scene synthesis for 3D environments has been a long standing challenge in computer graphics for many decades. In particular, indoor scenes have been of interest due to the wide range of applications in virtual and mixed reality, robotics, entertainment, retail, virtual staging, interior design, and more. As the manual creation and editing of such scenes requires substantial human effort and expertise, significant effort has been put on automating this process. With early approaches revolving around heuristics-based methods and procedural modeling (Qi et al., 2018; Weiss et al., 2018; Xu et al., 2002; Yu et al., 2011; Purkait et al., 2020; Fisher et al., 2015), recent effort has shifted towards deep generative models, i.e., learning the distribution of indoor scenes directly from data. For instance, autoregressive models (Paschalidou et al., 2021; Wang et al., 2018; Ritchie et al., 2019) learn to stochastically predict a sequence of objects for full scene generation or completion. Another line of work explores diffusion-based models for scene synthesis (Tang et al., 2024; Hu et al., 2024; Maillard et al., 2024) by learning how to gradually denoise object properties from random noise. Methods based on scene graphs have also been proposed, either by assuming a high-level scene graph as input or by generating a scene graph in a first stage and then obtaining object properties via diffusion (Zhai et al., 2024; Lin & Mu, 2024). With the recent advent of instruction-tuned Large Language Models

(LLMs), agent-based approaches have been pursued, relying primarily on the inherent world model of LLMs (Sun et al., 2024a; Çelen et al., 2024; Yang et al., 2024d; Feng et al., 2023). However, existing methods face several limitations: they simplify object semantics via one-hot class labels, ignore room boundaries or rely on fixed-resolution floor plans regardless of scene complexity, depend on zero-shot LLMs or external optimization algorithms given nascent Spatial Foundation Models, and lack direct natural language modification capabilities. Consequently, controllable scene synthesis and editing with rich semantic and geometric understanding remains a significant challenge.

We propose RESPACE, a novel framework leveraging natural language for intuitive scene synthesis and editing through text commands for object addition, removal, and swapping. We frame scene manipulation as a next-token prediction task, enabled by a structured scene representation (SSR) in a JSON format that represents spatial information (i.e., room boundaries, object semantics, and precise placement) explicitly, and train a specialized model, SG-LLM, for object addition. SG-LLM takes a short object prompt as input and performs single object placement as output. It is prompted by a zero-shot LLM that serves as the user interface, decomposing user instructions into atomic additions/removals. For removals, the zero-shot LLM handles these directly through SSR text editing. Full scene synthesis can be achieved via object lists generated by the zero-shot LLM that SG-LLM processes autoregressively. We decouple 3D asset selection from the SSR using a stochastic sampler that matches both size and semantics from an existing catalog. Example prompt-output pairs are shown in Fig. 2 alongside a summary. For evaluation, we introduce a voxelization-based metric capturing fine-grained geometric interactions beyond bounding boxes, quantifying more realistic placements such as chairs partially under tables (Fig. 4 (C)). After Supervised Fine-Tuning (SFT) on instructions, we use this metric—alongside other constraints—as verifiable reward for preference alignment on SG-LLM. Experiments on the 3D-FRONT (Fu et al., 2021a) dataset demonstrate a new state-of-the-art for object placement and superior human-perceived quality for full scene synthesis. Code and dataset will be made available. In summary, our contributions are as follows:

- We present RESPACE, a novel method for controllable indoor scene synthesis and editing, framing object addition and removal via next-token prediction and text-driven commands.

- We feature a dual-stage pipeline (SFT+GRPO) for object addition and are first to apply preference optimization with Reinforcement Learning with Verifiable Rewards (RLVR) on it, surpassing state-of-the-art while achieving strong results for full scene synthesis.

- We introduce a lightweight and interpretable structured scene representation with natural-language object descriptions and explicit numerical values for scene boundaries and object positioning, enabling direct editing and asset-agnostic deployment across 3D catalogs.

- We propose the Voxelization-Based Loss (VBL), a novel *evaluation* metric capturing fine-grained geometric interactions beyond 3D bounding boxes (e.g., between chair and table).

## 2 RELATED WORK

**3D Indoor Scene Synthesis.** Early scene synthesis approaches relied on heuristics and procedural modeling (Qi et al., 2018; Weiss et al., 2018; Xu et al., 2002; Yu et al., 2011; Purkait et al., 2020; Fisher et al., 2015). With deep learning's emergence, transformers (Ritchie et al., 2019; Wang et al., 2021; Paschalidou et al., 2021) and diffusion models (Tang et al., 2024; Hu et al., 2024; Wei et al., 2023; Maillard et al., 2024) gained prominence. Deep Priors (Wang et al., 2018) introduced CNN-based attribute prediction, while Fast&Flexible (Ritchie et al., 2019) developed a chained CNN pipeline conditionable on floor plan images. SceneFormer (Wang et al., 2021) proposed autoregressive transformers conditioned on floor plans and text descriptions, while ATISS (Paschalidou et al., 2021) pioneered treating scenes as unordered object sets. FOREST2SEQ (Sun et al., 2024b) explores ordering strategies for autoregressive synthesis to improve placement quality. Recent advances include diffusion-based approaches like DiffuScene (Tang et al., 2024), Mi-Diff (Hu et al., 2024) (supporting floor plan conditioning via PointNet features), PhyScene (Yang et al., 2024b) (focusing on physically interactable synthesis), and LEGO-Net (Wei et al., 2023) (via rearrangement). Alternative approaches generate unified scene representations: Text2Room (Höllein et al., 2023) extracts textured meshes from 2D models, DreamScene (Li et al., 2024) uses Gaussian-based text-to-3D generation, and Set-the-Scene (Cohen-Bar et al., 2023) enables controllable NeRF scenes. Human-centric approaches include MIME (Yi et al., 2023) and SUMMON (Ye et al., 2022), while scene graph methods (Dhamo et al., 2021; Lin & Mu, 2024; Zhai et al., 2023a; 2024; Luo et al., 2020) like InstructScene (Lin &

Table 1: Comparison of key properties across indoor scene synthesis methods.

| Method | Non-Rectangular Layouts | Explicit Object Semantics | Text-Driven Editing | Trained Placement | Asset Sampling |
|---|---|---|---|---|---|
| ATISS Paschalidou et al. (2021) | ✓ | ✗ | ✗ | ✓ | ✗ |
| Mi-Diff Hu et al. (2024) | ✓ | ✗ | ✗ | ✓ | ✗ |
| LayoutGPT Feng et al. (2023) | ✗ | ✓ | ✗ | ✗ | ✗ |
| LayoutVLM Sun et al. (2024a) | ✗ | ✓ | ✗ | ✗ | ✗ |
| InstructScene Lin & Mu (2024) | ✗ | ✗ | ✗ | ✓ | ✗ |
| Ctrl-Room Fang et al. (2025) | ✓ | ✗ | ✗ | ✓ | ✗ |
| SceneWeaver Yang et al. (2025) | ✗ | ✓ | ✗ | ✓ | ✗ |
| **ReSpace (ours)** | ✓ | ✓ | ✓ | ✓ | ✓ |

Mu, 2024) and EchoScene (Zhai et al., 2024) use intermediate graph representations. Despite these advances, most methods either generate unified representations limiting asset flexibility, focus on end-to-end synthesis without granular editing capabilities, or lack explicit 3D boundary handling for complex layouts and intuitive text-driven manipulation.

**Language-based Scene Synthesis.** Early work like CLIP-Layout (Liu et al., 2023) explored text-prompted synthesis using CLIP (Radford et al., 2021) embeddings. With instruction-tuned LLMs, agent-based approaches evolved: LayoutGPT (Feng et al., 2023) pioneered zero-shot placement via CSS-based representation, while I-Design (Çelen et al., 2024), Holodeck (Yang et al., 2024d), and Open-Universe (Aguina-Kang et al., 2024) employ multi-agent systems to construct scene graphs or DSL instances before separate layout optimization. LayoutVLM (Sun et al., 2024a) generates text-based layouts with constraints before optimization, LLPlace (Yang et al., 2024c) retrieves assets via text prompts before using a fine-tuned LLM for placement, and SceneCraft (Kumaran et al., 2023) targets scene generation via iterative code generation with visual feedback. SceneWeaver (Yang et al., 2025) uses an LLM-based agent framework for text-driven scene synthesis. CASAGPT (Feng et al., 2025) targets cuboid arrangement for interior design but lacks natural language object descriptions and editing capabilities. More recently, Ctrl-Room (Fang et al., 2025) separates layout and appearance generation via a two-stage pipeline, achieving controllable text-to-3D room generation with mask-guided editing capabilities. However, these methods either require separate optimization stages, focus on open-domain generation, or lack explicit natural language semantics for object descriptions. Table 1 summarizes key properties across recent methods. Unlike prior work, our approach uses a specialized LLM for targeted indoor scene synthesis, directly predicting object semantics and positioning in a single forward pass while supporting probabilistic asset sampling. This remains fully generative (Bucher et al., 2023) while extending beyond rectangular floor plans to non-convex geometries.

**Preference Alignment and Test-Time Compute Scaling.** LLM development has evolved from pre-training only (GPT-3 (Brown et al., 2020) era) to dual-stage pipelines with instruction-tuning and preference alignment. Nominal works include InstructGPT (Ouyang et al., 2022), FLAN (Wei et al., 2021), Reinforcement Learning from Human Feedback (RLHF) (Christiano et al., 2017), Direct Preference Optimization (DPO) (Rafailov et al., 2023), and, most recently, Group Relative Policy Optimization (GRPO) (Shao et al., 2024) and RLVR (Lambert et al., 2024; Su et al., 2025)—all aligning models to maximize (learned) rewards. Recent work also focuses on increased test-time compute (Snell et al., 2024) via self-consistency (Wang et al., 2022), Best-of-N sampling, and reward models (Brown et al., 2024). Formulating scene synthesis via language modeling, we are the first, to our knowledge, performing preference alignment on this task via verifiable rewards.

## 3    RESPACE

We introduce RESPACE, a method for autoregressive indoor scene generation and editing via natural language that sequentially adds and removes objects to empty or partial scenes (Fig. 2).

**Problem Statement.** Given a user instruction $u_i$ in natural language, our goal is to learn the conditional distribution $\hat{S}_i \sim p_\theta(S_i|u_i)$ with input scene $S_i$ and modified scene $\hat{S}_i$. Let $\mathcal{S} = \{S_1, S_2, ..., S_N\}$ be a collection of indoor scenes, where each scene $S_i = (T, \mathcal{B}, \mathcal{O})$ is composed by its room type $T \in \mathcal{T}$, room boundaries $\mathcal{B} = \{\mathcal{B}_{\text{top}}, \mathcal{B}_{\text{bottom}}\}$, and unordered set of objects

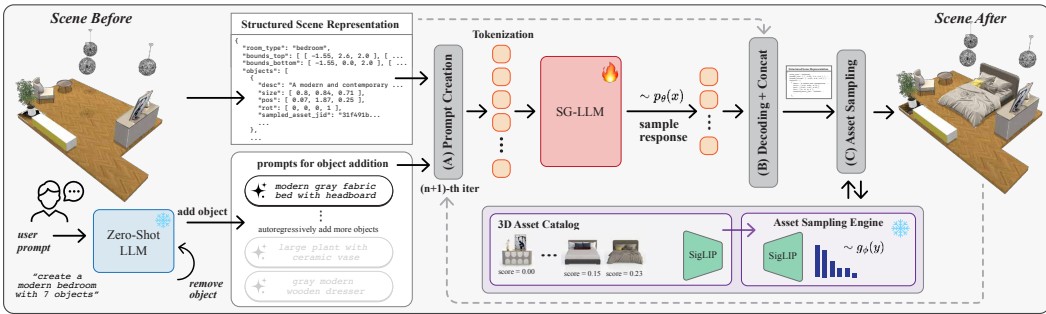

Figure 2: **ReSpace Overview.** Given a user instruction via text and an existing scene represented via SSR, we autoregressively perform 3D scene synthesis and editing. A zero-shot LLM converts user instructions to sequential commands for object removal and addition, with the latter done via specialized SG-LLM ($p_\theta$) and removal via zero-shot SSR editing.

$\mathcal{O} = \{O_1, O_2, ..., O_K\}$. Unlike previous work, our bounds are defined as ordered point sets $b_i \in \mathbb{R}^3$ forming closed rectilinear polygons—$\mathcal{B}_{\text{top}} = \{b_1, b_2, ...b_M\}$ for the ceiling and $\mathcal{B}_{\text{bottom}}$ for the floor. Further, each object in the scene $O_i = (d_i, h_i, t_i, r_i)$ is represented as a labeled 3D bounding box with asset description $d_i$, size $h_i \in \mathbb{R}^3$, position $t_i \in \mathbb{R}^3$, and orientation $r_i \in \mathbb{R}^4$. The object description $d_i$ captures fine-grained object semantics such as material, color, and style explicitly via text. Rotations are given as quaternions. We formulate the task of 3D scene synthesis as learning a generative model such that a scene with $K$ objects can be autoregressively composed from previously placed objects $\{O_{j<i}\}$, natural language prompt $p_i$, room boundaries $\mathcal{B}$, and room type $T$.

## 3.1 STRUCTURED SCENE REPRESENTATION

Given a scene $S_i = (T, \mathcal{B}, \mathcal{O})$, we propose a Structured Scene Representation (SSR) that follows a nested dictionary schema. This is inspired by hierarchical DSLs as seen in prior work on scene composition and shape programs (Zhang et al., 2024; Tian et al., 2019; Avetisyan et al., 2024), as well as structured representations in Structured3D (Zheng et al., 2020) and SpatialLM (Mao et al., 2025), but follows a simpler structure for 3D indoor scenes. Let the room type be given as a short text string, let boundaries $\mathcal{B}_{\text{top}}$ and $\mathcal{B}_{\text{bottom}}$ be given as a nested list of 3D coordinates, and let the set of objects be a flat list, with each object defined as a dictionary with its compact textual description, size, position, and rotation. A full example of our SSR is given in Appendix A.9, with a short snippet in Fig. 2. Note that the 3D asset choice is detached from the actual scene representation. Thus, SSR is an *abstraction* over any scene instance and allows to swap assets without changing the underlying SSR. This choice, in contrast to neural scene or voxel-based methods (Peng et al., 2020; Mildenhall et al., 2021), is lightweight ($\sim$ KBs), and directly editable. Further, it is extensible, e.g., by representing doors/windows or adding spatial relationships between objects for more fine-grained scene graphs.

## 3.2 SCENE SYNTHESIS VIA AUTOREGRESSIVE LANGUAGE MODELING

Given an SSR instance, we can tokenize a scene $S_i$ into $N$ text tokens $t_j$ such that $Tok(S_i) = \mathcal{U} = \{t_1, ..., t_N\}$. Let $\mathcal{U}_{prev}$ be the sequence of tokens for the existing scene that composes an SSR, and let $\mathcal{U}_i$ be the token sequence for the current object. Let $p_i$ represent the object prompt for the next object to add. Note that $\text{Tok}(S_i) = \mathcal{U}_{\text{prev}} + \mathcal{U}_i$, where the complete scene tokenization is the concatenation of the existing scene tokens and the new object tokens. We can formulate a generative model for autoregressive scene synthesis and completion as a conditional next-token prediction task:

$$p_\theta\big(O_i | p_i, \{O_{j<i}\}, T, \mathcal{B}\big) = p_\theta\big(\mathcal{U}_i | p_i, \mathcal{U}_{prev}\big) = \prod_{j=0}^{M} p_\theta\big(t_j | p_i, \mathcal{U}_{prev}, t_{<j}\big) \tag{1}$$

thus, during inference, sampling the next object for the scene involves sampling $M$ tokens from $p_\theta(\text{x})$ until the end-of-sequence (EOS) token is chosen. Let, $p_\theta(\text{x})$ be represented by an LLM and let this specially trained model for autoregressive object addition be denoted as SG-LLM (Scene Graph LLM). We show our pipeline in Fig. 2 for an example scene, where the full input string for SG-LLM is composed in step (A) from the existing SSR and a single object prompt. After tokenization, forward pass in the LLM, and response sampling, tokens get decoded and concatenated with the existing

| Input Image | Asset Description for SSR | Object Prompts for SG-LLM instructions |
|---|---|---|
| → GPT-4o → | *"Modern minimalist king-size bed with a wood frame, padded gray fabric headboard, and clean lines."* | → GPT-4o → *'bed'—'king-size bed'—'modern king bed'—'gray fabric bed'—'wood-framed king bed'—'minimalist king-size bed'—'padded headboard king bed'—'clean-lined wood frame bed'—'modern gray fabric headboard bed'—'minimalist padded gray headboard bed'* |

Figure 3: Example of description and prompt bank generation for a single asset in the catalog.

object list in step (B). Lastly, a 3D asset is sampled in step (C) via asset sampling engine. This process can be repeated $K$ times to iteratively add more objects, given $K$ object prompts.

**Stochastic Asset Sampling.** We can retrieve assets for added objects from a given 3D asset catalog using the descriptions and sizes of each object defined in the SSR. Prior work uses greedy selection via closest 3D bounding box match, filtered by class label (Feng et al., 2023; Paschalidou et al., 2021; Tang et al., 2024; Çelen et al., 2024; Yang et al., 2024c; Hu et al., 2024). In contrast, we formalize asset retrieval as a probabilistic process where each 3D asset mesh $m_i$ is drawn from a distribution parameterized by semantic and geometric constraints: $m_i \sim g_\phi(d_i, h_i)$, where $d_i$ is the natural language description and $h_i$ is the target size. The distribution $g_\phi$ computes scores as weighted combinations of semantic and geometric similarities: $\text{score}(m_j) = \lambda \cdot \text{sim}_{\text{sem}}(d_i, d_j) + (1-\lambda) \cdot \text{sim}_{\text{geo}}(h_i, h_j)$, where $\text{sim}_{\text{sem}}$ uses L2-normalized SigLIP (Zhai et al., 2023b) embeddings for text-to-asset matching and $\text{sim}_{\text{geo}}$ measures size compatibility via Gaussian similarity: $\exp(-\|s_i - s_j\|^2/(2\sigma^2))$. The final distribution is obtained through temperature-scaled softmax with nucleus sampling (top-$p$) and top-$k$ filtering. For deterministic 'greedy' retrieval, we can set $m_i = \text{argmax}_{m_j} g_\phi(d_i, h_i)$.

**Preference Alignment via GRPO.** With object addition formulated as language modeling, we leverage preference optimization with Group Relative Policy Optimization (GRPO) (Shao et al., 2024). Let for each iteration be $G$ candidates $a_i$ with verifiable reward $r_i$ and the objective:

$$J(\theta) = \frac{1}{G} \sum_{i=1}^{G} \big( \min \big( \frac{\pi_\theta(a_i|s)}{\pi_{old}(a_i|s)} A_i, \text{clip}(\frac{\pi_\theta(a_i|s)}{\pi_{old}(a_i|s)}, 1-\varepsilon, 1+\varepsilon) A_i \big) - \beta D_{KL}(\pi_\theta \| \pi_r) \big) \quad (2)$$

where each term in the sum is expanded as a per-token loss per response $a_i$. The advantage $A_i$ is given as $A_i = \frac{r_i - \text{mean}(r_1, \cdots, r_G)}{\text{std}(r_1, \cdots, r_G)}$, $\beta$ controls the KL divergence between the current policy $\pi_\theta$ and reference policy $\pi_r$, and $\varepsilon$ is given as upper/lower-bound for clipping.

**Full Scene Synthesis and Object Removal via Zero-Shot Learning.** Our method targets scene *editing* via autoregressive addition and removal, where removal is enabled through a zero-shot LLM (see Fig. 2 for an overview). When given a removal command, this LLM directly modifies the SSR by updating the JSON to remove specified objects. For full scene generation, we leverage this same zero-shot LLM's ($LLM_{ZS}$) inherent *world model* to generate an object prompt list $\mathcal{P}_i = \{p_1, ..., p_K\} \sim LLM_{ZS}(u_i)$ from user instruction $u_i$. A complete scene $S_i$ can then be created by autoregressively passing each object prompt for addition to our trained SG-LLM. System prompts for these components are reported in Appendix A.10. While using a single model for all scene synthesis and editing tasks would be theoretically preferable, this would require more complex fine-tuning in an SFT+RL pipeline to prevent mode collapse due to task/class imbalance. Thus, for this work, we focus on optimizing SG-LLM for object addition only, demonstrating the approach's viability while simplifying the training process by complementing it with a zero-shot LLM.

### 3.3 LAYOUT VIOLATIONS: VOXELIZATION-BASED LOSS

Representing scenes via SSR follows previous work in the sense that objects can be simplified by a collection of positioned 3D bounding boxes. We show a visualization in Fig. 4 (A), with blue boxes for objects and red small red cubes for ceiling and floor room bounds. Existing work on indoor scene synthesis does not study or report layout violations extensively; it reports 'out-of-bounds' as the ratio of scenes that have objects partially out-of-bounds (Çelen et al., 2024), or object-intersections via Intersection-over-Union (Hu et al., 2024) or average volume intersection (Çelen et al., 2024) on 3D bounding boxes. However, bounding-box-based metrics are unable to accurately evaluate realistic object placement (e.g., chair and table in Fig. 4 (A)) or provide an accurate signal of violations. We introduce the Voxelization-Based Loss (VBL), a novel geometry-aware evaluation metric that is defined as follows: First, we voxelize the scene boundary mesh $\mathcal{B}$ with fixed voxel size $G$ to create uniform grid $V_S$, and similarly voxelize each object mesh $O_j$ into binary occupancy grid

$\mathcal{V}_j \in \{0, 1\}^{x_j \times y_j \times z_j}$. To quantify layout violations, we define two metrics: (1) Out-of-Bounds Loss (OOB) counts voxels outside scene boundaries as $\text{OOB}_j = \sum_i \mathcal{V}_j(i) - \sum_i \mathcal{V}_j(i) \cdot \mathcal{V}_S(i)$ with total $\text{OOB} = \sum_j \text{OOB}_j$, and (2) Mesh Boundary Loss (MBL) measures voxel overlap between unique object pairs $(O_m, O_n)$ as $\text{MBL}_{(m,n)} = \sum_i \mathcal{V}_m(i) \cdot \mathcal{V}_n(i)$. Since MBL is symmetric, it is computed once per unique pair, with total $\text{MBL} = \sum_{m<n} \text{MBL}_{(m,n)}$. The total Voxel-Based Loss (VBL) is the sum of these two metrics, quantifying how many voxels 'violate' the layout. Fig. 4 shows these violations with OOB voxels in red and MBL voxels in purple. We motivate OOB and MBL as separate sub-metrics as they are complementary—objects outside boundaries have high OOB but low MBL since they are less likely to intersect with other objects. It is thus crucial to minimize both metrics. Since MBL scales subquadratically with object count, we implement a horizontal 2D intersection check for early stopping, skipping full computation when 2D projections show zero violations. We find that a voxel size of $G = 0.05m$ provides an optimal compute/accuracy trade-off.

## 4 EXPERIMENTS

We conduct experiments on two main tasks: (i) single object addition on partial scenes via prompt-based instructions and (ii) full scene synthesis. For ours, we also study the effectiveness of object removal and perform various ablations studies for further analysis.

**Scene-Prompt Dataset.** We partition 3D-FRONT (Fu et al., 2021a) into 'bed' (bedrooms), 'liv' (living rooms and dining rooms), and 'all' splits (with 6328/500/500, 3830/500/500, and 12055/500/500 train/val/test samples respectively) after filtering out noisy samples via our voxelization-based method. Since the dataset lacks textual descriptions, we use GPT-4o (Hurst et al., 2024) as a vision-language model to generate detailed descriptions $\mathbf{d_j}$ for each object from provided 3D-FUTURE (Fu et al., 2021b) renderings, following vision-based approaches for labeling (Raghu et al., 2023; Aguina-Kang et al., 2024). We then create 10 unique, concise prompts per object to form our prompt bank $\mathcal{P}(o)$ (see Fig. 3). The prompt bank provides data augmentation with varying levels of detail for the same object—for example, a bed might have prompts ranging from "bed" (1 word) to "modern king-size platform bed" (4 words). During training, we sample $p_i \sim \text{Unif}(\mathcal{P}(o))$ for each object, ensuring the model learns a robust prompt-to-object mapping across different prompt styles. During training, we dynamically generate instruction triples, and include empty rooms (10%), scenes with a final object placement missing (10%), and partial scenes with random number of existing objects (80%). We create fixed test sets with 500 instructions and corresponding object prompts using three random seeds. More details are in Appendix A.1.

### 4.1 EXPERIMENTAL SETTINGS

**Baselines and Implementation.** We report implementation details in Appendix A.2 and compare our method to existing state-of-the-art 3D scene synthesis approaches: (i) ATISS (Paschalidou et al., 2021), a transformer-based auto-regressive model that treats scenes as unordered set of objects; and (ii) MiDiffusion (Hu et al., 2024) (Mi-Diff), a mixed discrete-continuous diffusion model for scene synthesis. Both models take a $256 \times 256$ top-down orthographic projection of the floor plan as an input condition. We map the prompt from each instruction to the corresponding ground-truth class label. ATISS is already autoregressive, and single object-addition via one-hot class label conditioning

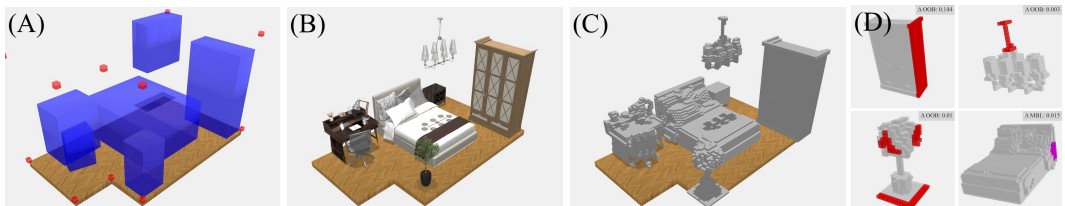

Figure 4: Scene represented with 3D bounding boxes in blue and bounds in red (A), with 3D assets (B), their voxelized counterpart (C), and some examples of OOB/MBL voxel violations (D). Note how the desk and chair interact smoothly in mesh space compared to their blue bounding boxes, while the lamp is largely OOB with its bounding box but only minor with its mesh.

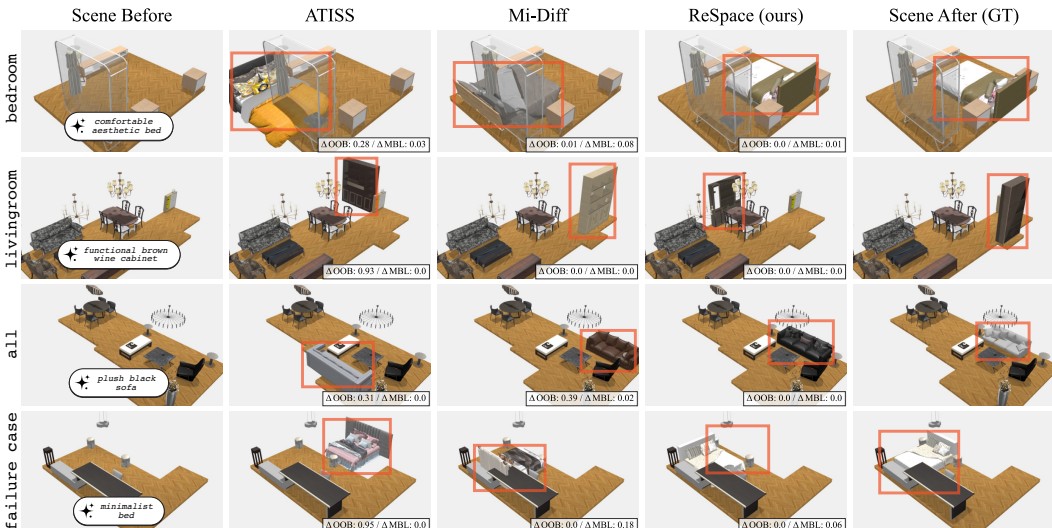

Figure 5: Qualitative results on single instructions, with our method performing the strongest. For ours, we use ReSpace/A$^\dagger$. We show a failure case on the last row where all methods perform poorly.

Table 2: Quantitative evaluation on **single-instruction object addition** on a hold-out test set of $3 \times 500$ fixed instructions and their corresponding object prompts, generated with 3 random seeds. Each method is trained on the same train splits. KID and Layout Violation (OOB, MBL, VBL) are multiplied by $10^3$ for readability. Best values are in **bold**, second best in underline.

| | Method | Scene Renderings | | | Layout Violations | | | Prompting |
|---|---|---|---|---|---|---|---|---|
| | | $\downarrow$ FID | $\downarrow$ FID$_{CLIP}$ | $\downarrow$ KID$_{\times 1e3}$ | $\downarrow$ OOB$^\Delta_{\times 1e3}$ | $\downarrow$ MBL$^\Delta_{\times 1e3}$ | $\downarrow$ VBL$^\Delta_{\times 1e3}$ | $\uparrow$ PMS |
| 'bed' | ATISS | $36.18_{\pm.3}$ | $1.74_{\pm.0}$ | $0.19_{\pm.0}$ | $97.70_{\pm6.0}$ | $13.54_{\pm0.5}$ | $111.24_{\pm5.4}$ | $0.58_{\pm.0}$ |
| | Mi-Diff | $36.12_{\pm.3}$ | $1.76_{\pm.1}$ | $\underline{0.05}_{\pm.0}$ | $64.04_{\pm5.3}$ | $14.27_{\pm1.5}$ | $78.31_{\pm4.1}$ | $0.57_{\pm.0}$ |
| | ReSpace/B | $\mathbf{35.10}_{\pm.3}$ | $\mathbf{1.66}_{\pm.0}$ | $-0.06_{\pm.1}$ | $\underline{20.68}_{\pm4.1}$ | $\mathbf{6.21}_{\pm1.0}$ | $\underline{26.89}_{\pm4.8}$ | $\underline{0.87}_{\pm.0}$ |
| | ReSpace/A$^\dagger$ | $\underline{35.51}_{\pm.2}$ | $\underline{1.67}_{\pm.0}$ | $0.06_{\pm.1}$ | $\mathbf{8.38}_{\pm1.5}$ | $\underline{8.13}_{\pm1.2}$ | $\mathbf{16.51}_{\pm3.0}$ | $\mathbf{0.89}_{\pm.0}$ |
| 'liv' | ATISS | $32.26_{\pm.1}$ | $1.48_{\pm.0}$ | $\underline{0.71}_{\pm.3}$ | $63.87_{\pm6.9}$ | $11.43_{\pm3.8}$ | $75.30_{\pm5.8}$ | $0.58_{\pm.0}$ |
| | Mi-Diff | $33.30_{\pm.3}$ | $1.53_{\pm.0}$ | $1.06_{\pm.2}$ | $43.88_{\pm7.6}$ | $12.87_{\pm1.4}$ | $56.75_{\pm8.8}$ | $0.56_{\pm.0}$ |
| | ReSpace/L | $\underline{31.94}_{\pm.1}$ | $\underline{1.41}_{\pm.0}$ | $0.24_{\pm.1}$ | $\underline{24.64}_{\pm7.8}$ | $\mathbf{7.36}_{\pm0.4}$ | $\underline{32.00}_{\pm7.8}$ | $\underline{0.84}_{\pm.0}$ |
| | ReSpace/A$^\dagger$ | $\mathbf{31.90}_{\pm.2}$ | $\mathbf{1.40}_{\pm.0}$ | $\mathbf{0.19}_{\pm.2}$ | $\mathbf{11.20}_{\pm3.1}$ | $\underline{8.22}_{\pm1.0}$ | $\mathbf{19.41}_{\pm4.1}$ | $\mathbf{0.87}_{\pm.0}$ |
| 'all' | ATISS | $36.40_{\pm.0}$ | $1.77_{\pm.0}$ | $0.22_{\pm.1}$ | $121.66_{\pm8.6}$ | $\underline{14.48}_{\pm1.0}$ | $136.14_{\pm8.7}$ | $\underline{0.57}_{\pm.0}$ |
| | Mi-Diff | $\underline{36.14}_{\pm.2}$ | $\underline{1.72}_{\pm.0}$ | $\underline{0.07}_{\pm.1}$ | $\underline{40.51}_{\pm5.5}$ | $18.19_{\pm0.6}$ | $\underline{58.70}_{\pm4.9}$ | $0.56_{\pm.0}$ |
| | ReSpace/A$^\dagger$ | $\mathbf{35.71}_{\pm.4}$ | $\mathbf{1.67}_{\pm.0}$ | $\mathbf{-0.03}_{\pm.1}$ | $\mathbf{13.11}_{\pm3.7}$ | $\mathbf{8.67}_{\pm2.3}$ | $\mathbf{21.78}_{\pm6.0}$ | $\mathbf{0.87}_{\pm.0}$ |

is natively supported. For Mi-Diff, we follow their masking strategy and enable de-noising only for a single object. We chose ATISS and Mi-Diff as they represent strong end-to-end trained methods that can be conditioned on non-rectangular floor plans, making them suitable baselines for both conditional object addition and full scene synthesis on our dataset.

Additionally, we compare our method to recent LLM-based frameworks: LayoutGPT (Feng et al., 2023) and LayoutVLM (Sun et al., 2024a). Since these methods are limited to rectangular floor plans, we evaluate them on a filtered rectangular-only subset from the 'all' split for full scene synthesis only. Note that LayoutVLM requires a set of objects with bounding boxes as input, whereas our method generates the object list from scratch, representing a fundamental difference in the setup.

For fair comparison, we use greedy asset selection since baselines only support deterministic retrieval, consistent with LLM evaluation practices using lower temperature to ensure reproducible comparisons without sampling variance (Bucher & Martini, 2024). We show details on stochastic selection for ours in Appendix A.3.

**Evaluation Metrics.** We use our introduced Voxelization-Based Loss (VBL) (see Section 3.3) as the main evaluation metric, and follow previous work (Paschalidou et al., 2021; Lin & Mu, 2024; Hu et al., 2024) by rendering a top-down projection for each scene, computing Fréchet Inception Distance (FID) (Heusel et al., 2017), $\text{FID}_{\text{CLIP}}$ (Kynkäänniemi et al.), and Kernel Inception Distance (KID) (Bińkowski et al., 2018) between train split and generated scenes. For train splits, we compute a set of $\min(N, 5000)$ renderings for full scenes and instructions (partial scenes) respectively. We also report Prompt Matching Score (PMS) to measure how many words $w_j$ from the prompt $p_i$ are captured via the description $d_i$ of the *sampled* 3D asset: $\text{PMS}(p_i, d_i) = \frac{1}{|p_i|} \sum_{w_j \in p_i} \mathbb{1}_{w_j \in d_i}$, with higher recall indicating better instruction-following capabilities. We use the postfixes '/B', '/L', and '/A' to denote the training set for SG-LLM, and denote with ReSpace/A$^{\dagger}$ the model with +GRPO.

## 4.2 Prompt-Driven Scene Editing

**Object Addition**. We present results for object addition (placing a single object into an empty or partial scene) in Table 2, reporting the *delta* VBL to quantify layout changes after insertion. Our method consistently outperforms existing baselines on all reported metrics across all three datasets. Notably, the model trained on 'all' (+GRPO) performs stronger even on 'bed' and 'liv' subsets, indicating that more diverse training scenes and our preference alignment method effectively boost performance, though the latter does not consistently outperform the SFT baseline on the MBL metric. Fig. 5 shows qualitative results, with our method performing strongly, sometimes exceeding ground-truth placements (e.g., third row, where the black sofa better matches the prompt and aligns well between side tables). The last row demonstrates a failure case with a bed placed outside scene boundaries for our method as well. Our method has significantly stronger performance on OOB metrics, empirically validating that an explicit boundary representation seems superior to fixed-resolution floor plan renderings as layout complexity beyond rectangular floor plans increases.

**Object Removal**. We additionally experiment with object removal using the same instructions from the test set. For this, we re-merge the intended object for addition (ground-truth) with the existing scene and prompt the system to *remove* the object given solely the object prompt. As scenes can contain multiple objects of the same asset, they will share the same description $d_i$. This can lead to ambiguity if only given a short object-level prompt. We treat a removal as correct if all assets matching the prompt 1:1 were removed from the scene and report accuracy as (# correct/# all), with $90.9\% \pm 0.6$ on 'bed', $75.2\% \pm 1.0$ on 'liv', and $87.3\% \pm 0.7$ on 'all'.

## 4.3 Prompt-Driven Full Scene Synthesis

Beyond object-level editing, we evaluate full scene synthesis, where our approach faces a fundamental challenge: unlike end-to-end trained baselines, our method relies on prompts from a zero-shot LLM. We inform the LLM with three priors: (1) the complete set of 3D-FRONT object classes, (2) a floor area to object count histogram for appropriate density sampling, and (3) few-shot examples of prompt lists from similar training scenes. While Table 3 shows mixed quantitative results with slightly higher FID/KID scores, comprehensive human evaluation with 125 participants performing 25,000 pairwise comparisons reveals ReSpace achieves 50.3%-54.2% win rates compared to 38.6%-41.9% for baselines. Our approach faces two primary limitations: the zero-shot LLM still acts as a bottleneck since it was not trained for this task, and we fix object counts based on floor area without adapting prompts after placement. Nevertheless, our emphasis on structured reasoning produces scenes that human evaluators consistently prefer, validating our design choices and demonstrating that slightly higher distribution divergence from the 3D-FRONT training data actually benefits realism. Fig. 6 shows qualitative samples with diverse outputs across methods, with more in Fig. 13.

## 4.4 Discussion

**Scene Complexity.** We study the effect of room size and existing object count on addition by clustering the number of objects per scene and aggregating them into uniform bins. We show mean and variance of Delta VBL for each bin in Fig. 7 alongside floor area size in light gray. We argue that an ideal model has uniform performance across varying object count, floor area, and scene density, and can see that our method performs much stronger compared to the baselines in this regard.

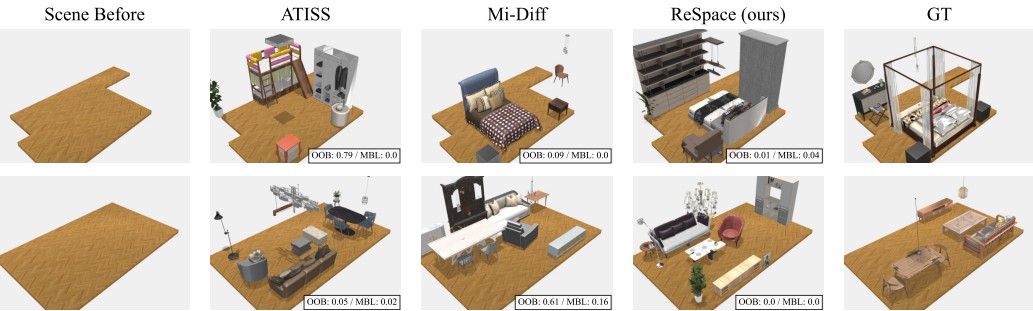

Figure 6: Qualitative results on full scene synthesis. For ours, we take results from ReSpace/A$^\dagger$.

Table 3: Quantitative evaluation on **full scenes** using 500 unseen floor plans with 3 random seeds per sample. Metrics follow Table 2. Human evaluation shows 50.3%-54.2% win rates vs 38.6%-41.9% for baselines (see Appendix A.5 for more details).

| | Method | Scene Renderings | | | Layout Violations | | | Prompt |
|---|---|---|---|---|---|---|---|---|
| | | ↓ FID | ↓ FID$_{CLIP}$ | ↓ KID$_{\times 1e3}$ | ↓ OOB$_{\times 1e3}$ | ↓ MBL$_{\times 1e3}$ | ↓ VBL$_{\times 1e3}$ | ↑ PMS |
| 'bed' | ATISS | $\underline{43.51}_{\pm.3}$ | $\underline{2.34}_{\pm.1}$ | $2.51_{\pm.5}$ | $414.3_{\pm23.6}$ | $\underline{99.7}_{\pm6.4}$ | $514.1_{\pm24.4}$ | n/a |
| | Mi-Diff | $\mathbf{43.18}_{\pm.4}$ | $\mathbf{2.23}_{\pm.1}$ | $\mathbf{1.34}_{\pm.2}$ | $\underline{360.1}_{\pm18.0}$ | $\mathbf{66.6}_{\pm9.5}$ | $\underline{427.0}_{\pm08.5}$ | n/a |
| | ReSpace/A$^\dagger$ | $44.77_{\pm.2}$ | $2.70_{\pm.0}$ | $2.17_{\pm.1}$ | $\mathbf{67.4}_{\pm07.1}$ | $140.7_{\pm20}$ | $\mathbf{208.1}_{\pm13.4}$ | $0.72_{\pm.0}$ |
| 'liv' | ATISS | $\underline{44.14}_{\pm.3}$ | $\underline{2.26}_{\pm.0}$ | $8.06_{\pm.3}$ | $506.6_{\pm22.2}$ | $\underline{135.1}_{\pm6.1}$ | $641.6_{\pm28.2}$ | n/a |
| | Mi-Diff | $\mathbf{40.76}_{\pm.1}$ | $\mathbf{2.11}_{\pm.1}$ | $\mathbf{4.29}_{\pm.1}$ | $\underline{361.5}_{\pm12.7}$ | $117.1_{\pm3.2}$ | $\mathbf{478.7}_{\pm09.6}$ | n/a |
| | ReSpace/A$^\dagger$ | $46.17_{\pm.3}$ | $2.42_{\pm.1}$ | $8.05_{\pm.7}$ | $\mathbf{254.4}_{\pm14.2}$ | $310.4_{\pm13}$ | $\underline{564.8}_{\pm25.8}$ | $0.73_{\pm.0}$ |
| 'all' | ATISS | $45.58_{\pm.1}$ | $\underline{2.37}_{\pm.0}$ | $3.87_{\pm.1}$ | $631.4_{\pm12.9}$ | $\underline{108.5}_{\pm6.9}$ | $739.8_{\pm19.0}$ | n/a |
| | Mi-Diff | $\mathbf{42.57}_{\pm.3}$ | $\mathbf{2.14}_{\pm.0}$ | $\mathbf{1.27}_{\pm.2}$ | $\underline{327.4}_{\pm41.3}$ | $\mathbf{87.1}_{\pm2.7}$ | $\underline{414.5}_{\pm41.6}$ | n/a |
| | ReSpace/A$^\dagger$ | $\underline{44.85}_{\pm.2}$ | $2.43_{\pm.2}$ | $\underline{2.44}_{\pm.5}$ | $\mathbf{160.2}_{\pm16.0}$ | $181.6_{\pm26.}$ | $\mathbf{341.8}_{\pm17.9}$ | $0.71_{\pm.0}$ |
| | LayoutGPT | $106.75_{\pm.5}$ | $9.17_{\pm.1}$ | $38.97_{\pm1.}$ | $1199.7_{\pm57.6}$ | $\mathbf{84.21}_{\pm06.0}$ | $1284.0_{\pm63.3}$ | n/a |
| | LayoutVLM | $80.04_{\pm.6}$ | $5.91_{\pm.1}$ | $6.33_{\pm.4}$ | $\mathbf{78.6}_{\pm02.2}$ | $84.34_{\pm03.6}$ | $\mathbf{162.9}_{\pm05.2}$ | n/a |
| | ReSpace/A$^\dagger$ | $\mathbf{70.86}_{\pm.9}$ | $\mathbf{4.57}_{\pm.2}$ | $\mathbf{2.07}_{\pm.0}$ | $88.8_{\pm07.4}$ | $124.60_{\pm23.6}$ | $213.4_{\pm24.3}$ | $0.72_{\pm.0}$ |

**Prompt Complexity.** We take the prompts for full scene synthesis, aggregate them by word count and report PMS for each bin, shown in Fig. 8. Longer prompts involve more complex instructions with more constraints (e.g., SG-LLM needs to satisfy color, material, shape, and style all at the same time). Thus, a negative correlation is confirmed by decreasing PMS with longer word count. We show PMS against number of objects on the same plot with separate x-axis, with slight negative correlation on increased object count as well. This shows that once the scene gets filled up with more objects, it seems harder for SG-LLM to satisfy semantic constraints.

**Scaling Test-Time Compute.** We explore Best-of-N (BoN) sampling to demonstrate that SG-LLM captures high-quality placement capabilities that are not consistently sampled. For each object

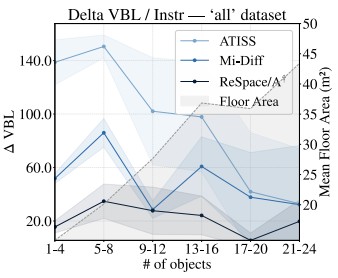

Figure 7: Delta VBL vs. # of objects. Ours is superior with more uniform performance.

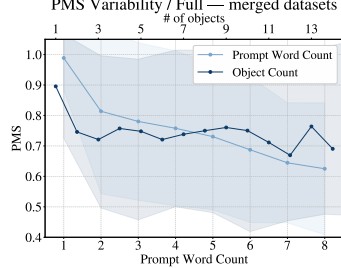

Figure 8: PMS vs. prompt word and object count, with (slight) negative correlation for both.

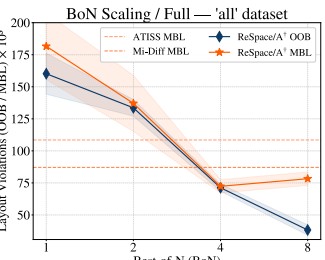

Figure 9: Test-Time Scaling via BoN, showing we can outperform baselines on MBL too.

addition, we sample $N$ responses and select the optimal one by filtering by highest PMS and choosing the one with lowest delta VBL. Fig. 9 shows results for $N = \{1, 2, 4, 8\}$ on the 'all' split. Both OOB and VBL decrease with a larger pool, eventually *surpassing* state-of-the-art for MBL on full scene synthesis. The effectiveness of BoN serves as evidence that although GRPO improves upon SFT-only for single object addition, the probability for sampling strong candidates could still be increased, indicating potential for further advances in preference alignment.

**VBL Evaluation and Asset-Agnostic Spatial Reasoning.** We emphasize that we use VBL primarily as *evaluation* metric and only as binary reward filter during GRPO, since we cannot backpropagate through it. While VBL rarely differs significantly from bounding box metrics, our strong human evaluation results confirm that VBL-based model selection corresponds to superior perceived scene quality. This suggests that semantic categories encode sufficient spatial constraints for effective scene arrangement, even without explicit geometric modeling, and validates a key hypothesis: SG-LLM can learn effective spatial reasoning purely from structured text, enabling deployment across arbitrary asset catalogs without requiring visual input or mesh-based geometry—critical for real-world scenarios where available physical inventory may vary and asset catalogs can be swapped.

**Limitations.** ReSpace faces limitations: (1) autoregressive placement can lead to suboptimal sequences where early object placement constrains later additions; (2) current focus on single-room scenes; (3) dependence on zero-shot LLM quality for complex instruction decomposition; (4) restriction to furniture objects rather than architectural elements like doors/windows; (5) potential for cluttered scenes when the object list provided via zero-shot LLM is too large; (6) editing operations limited to addition and removal.

## 5 CONCLUSION

We introduced RESPACE, a framework for text-driven 3D indoor scene synthesis using autoregressive language models. Our structured scene representation encodes explicit boundaries and positioning alongside textual descriptions for objects, while our specialized SG-LLM, trained via SFT+GRPO, surpasses state-of-the-art on object addition. By leveraging a zero-shot LLM for object removal and prompt generation, we demonstrate superior human-perceived scene quality for full scene synthesis without end-to-end training. This paradigm opens several promising research directions: developing a single model that handles all scene synthesis tasks while maintaining prompt-following capabilities; exploring scaling laws with larger context windows and model size as more training data can be made available; incorporating local correction via optimization after each autoregressive placement to eliminate layout violations while preserving generative diversity; and investigating advanced test-time compute techniques like Monte Carlo Tree Search to optimize full scene synthesis using PMS and VBL rewards directly, exploring alternative placement sequences through removal actions and branching instead of purely greedy addition.

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

# A APPENDIX

This Appendix provides the following:

- Details on the preprocessing of the 3D-FRONT dataset, instruction generation, and the creation of our SSR-3DFRONT dataset (Section A.1),
- Implementation details for training our SG-LLM (Section A.2),
- Details on stochastic asset sampling with qualitative results (Section A.3),
- Additional qualitative samples for full scene generation (Section A.4),
- Example of full SSR instance (Section A.9), and
- Prompts for zero-shot LLM for command decomposition and object removal (A.10).

## A.1 DATASET PREPROCESSING FOR SSR-3DFRONT

Our training data curation is based on the 3D-FRONT Fu et al. (2021a) dataset, which includes ∼19K synthetic indoor scenes of varying size and density, alongside positioned objects referenced from 3D-FUTURE Fu et al. (2021b), a furniture asset catalog providing textured 3D meshes and renderings for each asset. In order to bring the scenes from the 3D-FRONT dataset Fu et al. (2021a) into our Structured Scene Representation (SSR), we proceed as follows: First, we leverage an existing dataset of postprocessed 3D room meshes from 3D-FRONT from Dahnert et al. (2021) that has simplified wall geometry and closed holes, which facilitates the room boundary extraction. We run a custom line search algorithm (pseudo-code shown in Algorithm 1) on each room mesh to extract an ordered set of 3D vertices forming a rectilinear polygon for the floor and ceiling. These set of vertices build the room boundaries $\mathcal{B}_{\text{top}}$ and $\mathcal{B}_{\text{bottom}}$. Next, we convert the scenes into JSON, resembling our proposed SSR (see Section 3.1). We define 3 room types: `bedroom` for bedrooms, `livingroom` for living rooms, dining rooms and living/dining rooms, and `other` for all remaining rooms. We only consider scenes that have valid scene boundaries with $|\{b_i\}| \geq 4$, an object count of $3 \leq |\{o_i\}| \leq 50$, and VBL $< 0.1$. The latter filters out "invalid" scenes that contain too many object or boundary collisions (see Section 3.3). Since a few scenes have only a single malpositioned object, we further check for scene validity if a single object already violates this filter (via VBL $\geq 0.1$), and keep the modified scene if removing that object makes the scene valid. We shift all scenes to the origin $(0, 0, 0)$. In total, this results in 13055 valid scenes after preprocessing. "SSR-3DFRONT" is available via huggingface.co/datasets/gradient-spaces/SSR-3DFRONT

---

**Algorithm 1** Rectilinear Polygon Corner Extraction

---

**Require:** Mesh vertices $V \in \mathbb{R}^{n \times 3}$
**Ensure:** Corner vertices $C$ forming rectilinear polygon
1: $P \leftarrow \text{unique}(V[:, [0, 1]])$             ▷ Project to unique 2D points
2: $(x_0, y_0) \leftarrow \arg\min_{v:v_x=\min(P[:,0])} v_y$, dir $\leftarrow$ 'north', curr $\leftarrow (x_0, y_0)$, $C \leftarrow []$
3: **repeat**
4:      $S \leftarrow \text{getSortedAxisPoints}(P, \text{curr}, \text{dir})$      ▷ Points on same axis, sorted by direction
5:      $i \leftarrow \text{indexOf}(\text{curr}, S) + 1$
6:      **while** $\neg\text{isCornerPoint}(P, S[i], \text{dir})$ **do** $i \leftarrow i + 1$
7:      **end while**
8:      curr $\leftarrow S[i]$, dir $\leftarrow \text{getNextDirection}(P, \text{curr}, \text{dir})$, $C.\text{append}(\text{curr})$
9: **until** curr $= (x_0, y_0)$
10: **return** $C$

---

### A.1.1 PROMPTS FOR ASSET DESCRIPTION AND PROMPT BANK

As mentioned in Section 4, the raw 3D-FRONT dataset does not contain textual descriptions of assets or scenes, and we leverage GPT-4o Hurst et al. (2024) as a Vision-Language Model (VLM) to generate sentence-level descriptions $\mathbf{d_j}$ for each object $\mathbf{o_j}$ in the catalog. We query the VLM by attaching a rendering of the asset with a prompt that includes the provided class label. After obtaining sentence-level descriptions for each generation, we leverage the same VLM to generate 10 unique, concise prompts (2-5 words in noun phrase format) for each asset description $d_j$. This

```
User Prompt
Please provide a concise JSON object of the furniture item in the image using 'style', 'color',
'material', 'characteristics', and 'summary' as keys. Describe the style, noting any blends of
design elements. Specify the materials used for different components (if applicable). List the key
characteristics, including the shape, design features, and any distinctive elements or decorative accents.
If there are multiple values for a key, use a list of strings. DO NOT build a nested JSON. The summary
compactly captures the essence of the furniture's style, functionality, and aesthetic appeal, emphasizing
its unique attributes. This description should clearly differentiate this piece from others while
succinctly capturing its essential properties and we will use it for object retrieval, so it should be
as accurate as possible, keyword-heavy, but just be one extremely short sentence. You are an interior
designer EXPERT. Hint: It's a {ASSET_OBJECT_CATEGORY_LABEL}. Only output the JSON as a plain string and
nothing else.
```

Figure 10: Prompt for GPT-4o Hurst et al. (2024) for extracting various object properties for each asset including a sentence-level asset description, given a rendering of an object in the 3D-FUTURE dataset Fu et al. (2021b).

```
User Prompt
The list below contains a sentence referring to a single piece of furniture. Your task it to create a
list of 10 short descriptions that vary in length. Each description refers to the subject with a maximum
of 3-4 additional descriptive words that reference the color, style, shape, etc. All your sentences
should be in 'noun phrase'. You MUST include a variety of lengths in your descriptions, ensuring a few
samples are very short (1-2 words max) and others are longer (4-5 words). Have at least one sample with
only one word, except if you need to be more specific for the subject, e.g., use 'Coffee Table', not
just 'Table', if present. Use mostly basic properties such as color or material, but also include a few
creative and diverse versions to increase robustness in our ML training dataset.
The sentence is:
- {ASSET_DESCRIPTION}
Just output a plain list and nothing else. You have only one list of 10 descriptions. You MUST always
point to the referenced object above and not hallucinate other furniture or be overly generic by using
'furniture' or 'piece'. Every list contains the descriptions in increasing word length. Just output the
final JSON object as a plain string without any key. Never use markdown or ```json.
```

Figure 11: Prompt for GPT-4o Hurst et al. (2024) for generating a prompt bank with a list of 10 unique prompts, given a sentence-level asset description.

approach serves two purposes: (i) it covers diverse prompting styles with varying levels of detail and word order, and (ii) it prevents trivial overfitting on our small dataset by avoiding repetition of identical prompts that lead to memorization rather than generalization. We provide the full prompt for extracting visual properties in textual form for each asset in the 3D-FUTURE dataset Fu et al. (2021b) in Figure 10 and for prompt generation in Figure11. For asset descriptions, we leverage the content provided in the 'summary' as it seemed to best capture dense semantics that refer to style, color, material, etc.

### A.1.2 INSTRUCTION GENERATION FOR SG-LLM

Given the full scenes, we impose dynamic instruction generations based on a stochastic recipe. Let $\mathcal{P}(o) = \{p_1, \ldots, p_K\}$ be the fixed prompt bank for object $o$ (we set $K = 10$). During training, we turn $\mathcal{S}$ into an instruction tuple: $\mathcal{I} = (\hat{\mathcal{S}}, p, o_{\text{add}})$, where the model must learn to add object $o_{\text{add}}$ to the partial scene $\hat{\mathcal{S}} = (\mathcal{T}, \mathcal{B}, \hat{\mathcal{O}})$ when conditioned on the natural-language prompt $p$. To generate a tuple, we first draw a random permutation for the order of objects $\pi \sim \text{Unif}(S_N)$, then uniformly sample the prompt $p \sim \text{Unif}(\mathcal{P}(o_{\text{add}}))$ for object $o_{\text{add}}$. Let $Z \in \{Z_0, Z_1, Z_2\}$ be the instruction type: $Z_0$ ('zero_start'), $Z_1$ ('full_scene'), and $Z_2$ ('random'), with $Z_0$ teaching the model to start from an empty room given only the prompt, $Z_1$ teaching 'scene completion' as the final contains all objects from the scene but $o_{\text{add}}$, and $Z_2$ teaching robust object placements on arbitrary, shuffled partial scenes. For $Z_0$ we set $\hat{\mathcal{O}} = \varnothing$, $o_{\text{add}} = o_{\pi(1)}$. For $Z_1$ we set $\hat{\mathcal{O}} = \{o_{\pi(1)}, \ldots, o_{\pi(N-1)}\}$, $o_{\text{add}} = o_{\pi(N)}$. For $Z_2$ we draw a drop count $M \sim \text{Unif}\{0, \ldots, N-1\}$, put $L = N - M$ and define $o_{\text{add}} = o_{\pi(L)}$, $\hat{\mathcal{O}} = \{o_{\pi(1)}, \ldots, o_{\pi(L-1)}\}$. Instruction type is sampled as $Z \sim \text{Cat}(w_0, w_1, w_2)$ with fixed $w_0 = w_1 = 0.1$, $w_2 = 0.8$ and the conditional distribution factorizes as

$$p(\mathcal{I} \mid \mathcal{S}) = \sum_{z=0}^{2} w_z \, p(\mathcal{I} \mid Z = z, \mathcal{S}),$$

$$p(\mathcal{I} \mid Z = z, \mathcal{S}) = \begin{cases} \dfrac{\mathbf{1}_{\{z=0\}}}{N \, |\mathcal{P}(o_{\text{add}})|} \ (z = 0), & \dfrac{\mathbf{1}_{\{z=1\}}}{N! \, |\mathcal{P}(o_{\text{add}})|} \ (z = 1), & \dfrac{\mathbf{1}_{\{z=2\}}}{N! \, N \, |\mathcal{P}(o_{\text{add}})|} \ (z = 2) \end{cases}$$

Thus, for $Z_2$, we choose one of $N!$ permutations, one of $N$ drop counts, and one of $|\mathcal{P}(o_{\text{add}})|$ prompts. Fixed weights $w_0 = w_1 = 0.1$ guarantee at least 20% exposure to the empty-room and near-complete-room edge cases even for very large scenes. Since we have have empty or full scenes with $\frac{1}{N}$ probability (and partial scenes otherwise), their likelihood decreases inversely proportional with higher object count. Imposing minimum exposure via fixed weights ensures the model learns these edge cases as well. We perform random data augmentation on train/val samples by (i) rotating each scene by $\theta \in \{0, 90, 180, 270\}°$, (ii) cyclically shifting room bounds in a round-robin fashion, and (iii) slightly perturbing $x-$ and $z-$components of every position and size vector of each object with a uniform delta with $v' = v + \delta$ and $\delta \sim \mathrm{U}(-0.02, 0.02)$ for coordinate values $v$.

## A.2  IMPLEMENTATION DETAILS FOR SG-LLM

For Baselines, we use the released source code, modify the pre-processing to fit our custom dataset splits, and re-train Mi-Diff and ATISS from scratch on our three different datasets. We do not modify their hyperparameter choice and pick the best model based on their lowest validation loss.

We trained SG-LLM on a two-stage pipeline using 4xA100 NVIDIA 80GB GPUs with 16 CPUs and 384GB RAM, running Python 3.9.0 with CUDA 12.1.1, GCC 10.3.0, and `bf16` numerical precision. We conducted extensive experiments with 1B and 0.5B models, but observed best results with `Qwen2.5-1.5B-Instruct` (Yang et al., 2024a), which we use for all experiments together with `Llama-3.1-8B-Instruct` (Grattafiori et al., 2024) for the zero-shot LLM. For the first stage, we perform Supervised Fine-Tuning (SFT) on the full weights for 30-50 hours with a learning rate (LR) of 5e-5, local batch size of 4, gradient accumulation step (GAS) of 8, and a context window of 3000 tokens, selecting the model with best validation loss via mean delta VBL on the val split. We experimented with Low-Rank Adaptation (LoRA) Hu et al. (2022) but observed faster convergence with SFT on full weights. For GRPO fine-tuning, we use an LR of 5e-5, temp $= 0.7$, batch size of 4, GAS of 16, 4 generations per sample/instruction, and set $\beta = 0.0$ to cancel out the KL divergence term. We train for around 30 hours and select models based on best delta VBL, resulting in optimal models at around 70 epochs for SFT and 3 epochs for GRPO. For GRPO, we give rewards of $-1.0$ for invalid JSON outputs and $1.0$ for candidates $a_i$ that pass the filter: $\mathrm{PMS}(a_i) \geq 0.85$, $\mathrm{DSS}(a_i) \geq 0.9$, $\mathrm{VBL}(a_i) < 1e-5$, and $||(s_{a_i}^{GT} - s_{a_i})||^2 < 0.2$, where the latter is the L2 distance between predicted size $s_{a_i}$ and ground-truth (GT) size $s_{a_i}^{GT}$ of the 3D bounding box, and $\mathrm{DSS}(a_i)$ denotes the cosine similarity of the SigLIP embeddings between GT `desc` and predicted one. We further introduce a *high-quality-only* distillation that cancels out the loss for samples with valid JSON but not satisfying the above filter. We observed instability with higher LRs (>5e-5) and without excluding candidates with valid JSONs but object properties that do not pass the filter. Since high-quality samples only appear with around 25% probability, negative rewards dominate and corrupt the SFT behavior too aggressively (especially the JSON structure). Without our strong filter for high-quality samples, we observed reward hacking via GRPO: the model produced valid JSONs but optimized description and size such that smaller objects got placed, effectively reducing intersection probability and decreasing VBL. However, as this is not desired behavior, we imposed a stronger filter for positive rewards. Future work can investigate the trade-offs between LR, reward shaping, and reward distribution.

## A.3  STOCHASTIC ASSET SAMPLING

Our proposed stochastic asset sampling involves various hyperparameters to tweak the final discrete distribution. We heuristically found that $\lambda = 0.5$, $\sigma = 0.2$, temp $= 0.2$, top_p $= 0.95$, and top_k$=20$ perform the best. However, for all experiments reported in the results of the main paper (see Section 4), we impose a *greedy selection* strategy for asset sampling in order to maintain better comparison with baselines. We use the same hyperparameters as above and choose the top-1 asset via

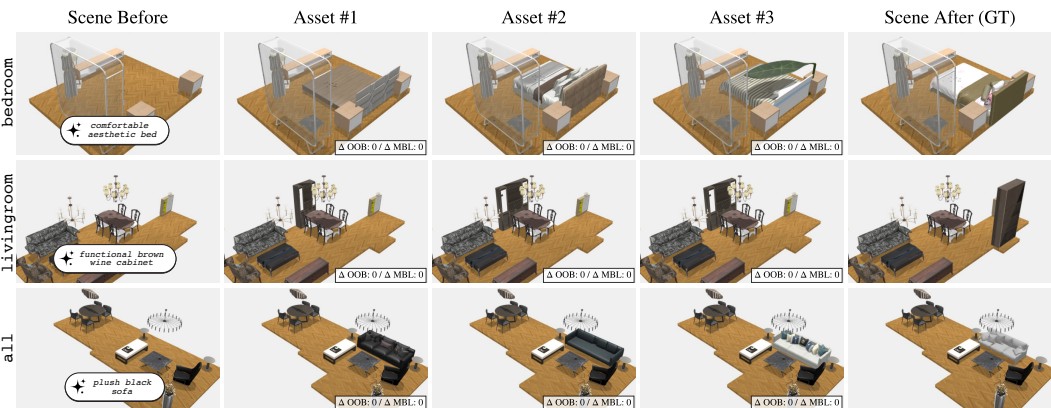

Figure 12: Qualitative results for true stochastic asset sampling without greedy selection (as done for the samples shown in Figure 5).

$\text{argmax}_{m_j} g_\phi(d_i, h_i)$. For comparison, we show true asset sampling (with the same hyperparameters) in Figure 12 on the same instructions from Figure 5 and 3 randomly sampled assets. We can simply sample from the distribution (instead of top-1 selection) for true stochasticity. We suggest that there might not be a single set of best hyperparameters for asset sampling. Instead, the user might tweak $\lambda$, (i.e., the strength of the semantic embedding), or $\sigma$ (i.e., the sharpness of the size matching via 3D bounding boxes) dynamically during scene generation to guide the process towards more desired candidates. With $\lambda = 0.5$, both geometry (via 3d bounding box size differences) and semantics (via SigLIP embeddings) have equal contribution to the final distribution for samples picked in Figure 12.

## A.4 MORE QUALITATIVE EXAMPLES ON FULL SCENES

In Figure 13, we show additional qualitative samples for full scene synthesis (with greedy asset sampling; otherwise same setup as done in the main experiments in Section 4). In contrast to Figure 6 where we showed the empty floor plan on the very left, we omit that column and replace it with Best-of-N (BoN) results from our method with BoN=8 on the very far right column. Due to the inherent randomness in our full pipeline between different BoN runs (i.e., especially involving the zero-shot LLM for the prompt list generation and the sampling of few shot samples and number of objects via priors), results for ours with BoN=1 and BoN=8 do not consistently involve the same object prompt lists into SG-LLM (i.e., 2nd column or 2nd to last column contain different scene-level compositions). Despite this fact, the BoN=8 results show superior performance compared to BoN=1, with less OOB and MBL, and better overall composition.

## A.5 USER STUDY 1: OURS VS. BASELINES

We conducted a comprehensive human evaluation study to validate whether our approach produces higher-quality scenes despite slightly elevated FID/KID scores compared to baselines. The study involved 125 participants from 24 nationalities performing 25,000 pairwise comparisons across 250 randomly sampled scenes generated via full scene synthesis. Participants were shown pairs of generated scenes and asked to select which appeared more realistic and well-arranged. We used Bradley-Terry analysis to rank the methods based on human preferences. The results demonstrate that both ReSpace variants significantly outperform end-to-end trained baselines in human-perceived quality, with win rates of 50.3% (ReSpace BoN=1) and 54.2% (ReSpace BoN=8) compared to 41.9% for Mi-Diff and 38.6% for ATISS. This empirical evidence validates our hypothesis that slightly higher distribution divergence from training data (as measured by FID/KID) can actually benefit scene diversity and realism as judged by human evaluators, and confirms that quantitative metrics may not fully capture human perceptions of scene quality.

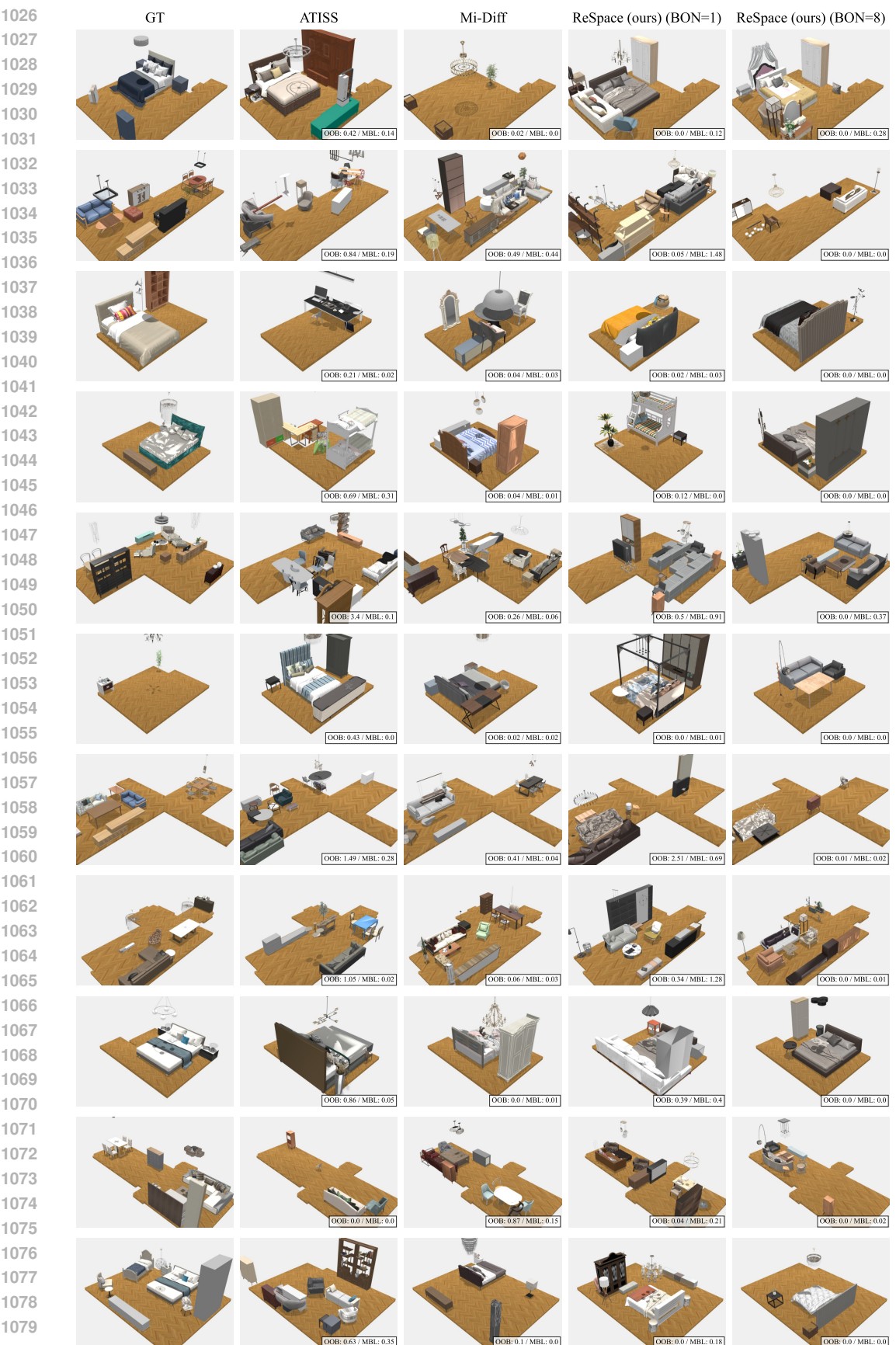

Figure 13: Qualitative results (random selection) on full scene synthesis compared with baselines. For ours, we show both BoN=1 and BoN=8.

Table 4: Human evaluation results using Bradley-Terry analysis on full scene synthesis. Results based on 25,000 pairwise comparisons from 125 participants across 250 randomly sampled scenes.

| Rank | Method | Bradley-Terry Score | Std Dev | Win Rate |
|------|--------|---------------------|---------|----------|
| 1 | Ground Truth | 0.3130 | 0.0045 | 64.9% |
| 2 | ReSpace (BoN=8) | 0.2186 | 0.0037 | **54.2%** |
| 3 | ReSpace (BoN=1) | 0.1923 | 0.0032 | **50.3%** |
| 4 | Mi-Diff | 0.1458 | 0.0025 | 41.9% |
| 5 | ATISS | 0.1304 | 0.0025 | 38.6% |

## A.6 USER STUDY 2: SFT VS. GRPO

To further validate whether our preference alignment approach (GRPO) influences human-perceived scene quality compared to supervised fine-tuning alone (SFT), we conducted a second targeted human evaluation study. This study involved 150 participants performing 4,500 pairwise comparisons specifically between scenes generated by our SFT-only model (ReSpace/A) and our SFT+GRPO model (ReSpace/A†) using the same prompts and floor plans. While the preference difference is modest and not statistically significant (51% vs 49% win rate), the results demonstrate that our preference alignment approach can positively influence human-perceived scene quality without degrading the underlying spatial reasoning capabilities learned during supervised fine-tuning. This finding is important given the training fragility we observed during GRPO, where aggressive preference optimization could corrupt the model's JSON generation and spatial reasoning abilities. The study confirms that our conservative approach to preference alignment preserves scene quality while providing directional improvements in layout violations as measured by our VBL metric.

Table 5: Human evaluation comparing SFT vs SFT+GRPO on full scene synthesis. Results based on 4,500 pairwise comparisons from 150 participants comparing identical prompts.

| Rank | Method | Bradley-Terry Score | Std Dev | Win Rate |
|------|--------|---------------------|---------|----------|
| 1 | ReSpace/A† (SFT+GRPO) | 0.5089 | 0.0076 | 51.0% |
| 2 | ReSpace/A (SFT only) | 0.4911 | 0.0076 | 49.0% |

## A.7 RUNTIME ANALYSIS (LATENCY FOR INFERENCE)

We compare the latency of our method with other baselines in order to get a better understand the design trade-offs. For this, we run full scene synthesis with $N = 50$ for each method on rectangular rooms from the 'all' test set and report mean and variance in seconds in Table 6.

Table 6: Runtime Analysis ours vs. baselines

| Rank | Method | Runtime (s) |
|------|--------|-------------|
| 1 | ATISS | $00.54_{\pm 00.17}$ |
| 2 | Mi-Diff | $03.79_{\pm 00.22}$ |
| 3 | ReSpace (BoN=1) | $08.70_{\pm 07.21}$ |
| 4 | LayoutGPT | $10.72_{\pm 04.33}$ |
| 5 | ReSpace (BoN=8) | $26.21_{\pm 25.22}$ |
| 6 | LayoutVLM | $42.89_{\pm 16.88}$ |

## A.8 REMOVAL OPERATION ANALYSIS

To investigate the relatively low removal accuracy of $75.2\% \pm 1.0$ on the 'liv' dataset compared to $90.9\% \pm 0.6$ on 'bed' and $87.3\% \pm 0.7$ on 'all', we conduct a detailed analysis across three dimensions: SSR length, prompt length, and failure modes.

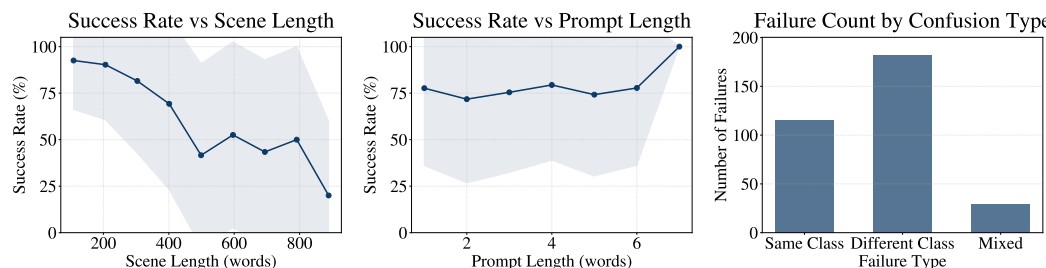

Figure 14: Analysis of removal accuracy on the 'liv' dataset.

**SSR Length Impact.** Figure A.6 (left) shows removal accuracy as a function of SSR word count. We observe a dramatic drop in performance: scenes with <200 words achieve 95% accuracy, while scenes with >500 words drop below 35%. This strongly confirms that longer token sequences present a fundamental challenge for the 8B instruction-tuned model (Llama-3.1-8B-Instruct) used for removal. The 'liv' split contains significantly more objects per scene, resulting in longer SSRs and explaining the performance gap. We hypothesize that larger instruction-tuned models (>70B) would better handle long-context JSON manipulation, though compute constraints in an academic lab prevented testing this hypothesis.

**Prompt Length and Ambiguity.** Figure A.6 (middle) examines accuracy versus object prompt length. Longer prompts (7 words) achieve 100% accuracy compared to ∼75% for shorter prompts, suggesting that more specific descriptions reduce ambiguity. However, accuracy remains relatively flat from 1-6 words (∼75%), indicating that prompt ambiguity can not be the only reason.

**Failure Mode Analysis.** Figure A.6 (right) categorizes the 326 total failures across the 'liv' test set. The primary bottleneck is *different class* errors (182 failures, 56%), where the model removes an object of the wrong class entirely (e.g., removing a table when prompted to remove a chair). This represents a fundamental reasoning failure rather than semantic ambiguity. We observe 115 *same class* failures (35%), where multiple objects of the requested class exist but the wrong instance is removed. This can be attributed to both prompt ambiguity and our evaluation methodology, which requires all objects with the same ground-truth 'desc' property to be removed for correctness. Finally, 29 failures (9%) involve removing both correct and incorrect objects simultaneously.

These results suggest that removal accuracy is primarily limited by (1) longer-context reasoning capabilities of the 8B model and (2) fundamental JSON manipulation errors, rather than just prompt ambiguity. Larger instruction-tuned models would likely address both issues, particularly the "different class" failures which represent over half of all errors.

A.9 EXAMPLE OF FULL SSR INSTANCE

We show a full example of a Structured Scene Representation (SSR) instance with sampled assets in Listing 1. The "abstract" SSR—before concrete 3D asset selection—would simply not contain the key/value pairs with 'sampled_' prefix and the optional 'uuid' key/value pair, as they are added after asset selection. For numerical values, we omit 'pretty formatting' (with line breaks after every element) in order to fit the example into a single page in this PDF.

Listing 1: Example of SSR instance with sampled assets

```
1  {
2    "room_type": "bedroom",
3    "bounds_top": [[-1.55, 2.6, 1.9], [1.55, 2.6, 1.9], [1.55, 2.6, -1.9],
        [-1.55, 2.6, -1.9]],
4    "bounds_bottom": [[-1.55, 0.0, 1.9], [1.55, 0.0, 1.9], [1.55, 0.0,
        -1.9], [-1.55, 0.0, -1.9]],
5    "objects": [
6      {
7        "desc": "A contemporary king-size bed with a brown padded headboard
            , Hello Kitty-themed pink and white bedding, graphic pillows,
            and bolster cushions, offering a comfortable aesthetic",
8        "size": [ 1.77, 0.99, 1.94 ],
```

```
 9        "pos": [ 0.44, 0.0, -0.44 ],
10        "rot": [ 0.0, 0.70711, 0.0, -0.70711 ],
11        "jid": "8a31d51c-2306-439f-90c6-650be7284975",
12        "sampled_asset_jid": "7bf721bf-8839-4343-95c5-b6e852805ad1",
13        "sampled_asset_desc": "Modern minimalist king-size bed with a wood
              frame, padded gray fabric headboard, and clean lines.",
14        "sampled_asset_size": [1.77, 1.02, 2.03],
15        "uuid": "d3d31dbc-ff1d-4122-8a80-52598c326f00"
16     }, ... ]
17 }
```

## A.10 PROMPTS FOR ZERO-SHOT MODEL

The full prompt for user instruction decomposition is in Figure 15. We further show an example of input/output prompts for a full scene generation in Figure 16. The prompt for object removal, using the same zero-shot LLM, is shown in Figure 17, with an example of input/output in Figure 18.

```
System Prompt
you are a world-class leading interior design expert. your task is to fulfill the request of the user
about interior design but you have help of another world-class expert model that can only be called in an
XML-style API.
# input
- <prompt> : the user request
- <scenegraph> : the current scene will be given as a JSON object. in some cases, there will be no scene
graph given, which means there is no "current" scene to work with. the "bounds_top" and "bounds_bottom"
keys contain the boundaries as a list of 3D vertices in metric space.
# task
- composing a list of commands to fulfill the user request via <add> and <remove> commands. ideally, you
reflect the existing objects in the scenegraph, if one is given.
# adding
- if the user wants to add one or multiple objects, you create an <add> command for every object/furniture
and add it to the list in "commands".
- for the description, you should refer to the subject with a maximum of five additional descriptive
words. the first words should refer to the color / style / shape / etc., while the last word should
always be the main subject. your description must be in 'noun phrase'.
- if the user request provides an existing scene description provided via <scenegraph>...</scenegraph>
and there are existing objects in the scene, you should try to match the style of the existing objects by
providing a similar style as part of the description of your commands.
- if the user provides some requirement about particular furniture that should be present in the room, you
should always add these objects via <add> commands.
- your format should be: <add>description</add>
- DO NEVER use more than 5 words for each description
# removing / swapping
- if the user wants to remove one to multiple objects, you add a <remove> command for every object that
should be removed.
- if the user wants to swap or replace furniture, you MUST use <remove> first and then use <add>
- if there are similar candidates for removal you should remove the object that matches the description
best.
- your format should be: <remove>description</remove>
- you can keep the description short here as well
# output
- the commands are given as a list under the "commands" key where each command follows EXACTLY the format
specified above and is given as a string, i.e. "<add>...</add>" or "<remove>...</remove>".
- if there are remove commands, you always put them BEFORE add commands.
- IMPORTANT: you NEVER use the <remove> commands unless the user EXPLICITLY asks for it via swapping or
removing objects. you do not make assumptions about this.
- you NEVER remove objects to "match the style" or if there is already an object in the scene similar
to the requested one. a scene can contain as many similar objects as the user wants. you ONLY remove
objects if the user explicitly asks for removal or swapping.
- if you use the <remove> command, you MUST provide your reasoning under the "reasoning" key, which comes
before the "commands" key in the same JSON object. - you always output the final JSON object as a plain
string and nothing else. NEVER use markdown.
# available object classes
- you should only pick objects for <add> based on the following high-level abstract classes
- your objects should be more specific than these classes but you should not add objects that are not part
of these classes/labels
{UNIQUE_OBJECT_CLASSES}
# available object classes
- you should only pick objects for <add> based on the following high-level abstract classes
- your objects should be more specific than these classes but you should not add objects that are not part
of these classes/labels
# few-shot examples for scenes that have a similar size to the requested one (your scene should be
different though and stick to the user prompt):{PROMPT_LISTS_FOR_K_EXAMPLES}
REMINDER: each description in your <add>...</add> commands should be IN NOUN PHRASE WITH 2-3 WORDS AND AT
MAXIMUM 5 words
```

```
User Prompt
<prompt>{UNSTRUCTURED_USER_INSTRUCTION}<prompt>
<scenegraph>{JSON_DUMP_OF_SSR_IF_PROVIDED_OR_NONE}</scenegraph>
```

Figure 15: System and User Prompt for the zero-shot LLM for command decomposition.

```
User Prompt
<prompt>create a bedroom with 5 objects.<prompt> <scenegraph>"room_type": "bedroom", "bounds_top":
[[-3.2, 2.65, 1.7], [3.2, 2.65, 1.7], [3.2, 2.65, -0.1], [0.9, 2.65, -0.1], [0.9, 2.65, -1.7], [-3.2,
2.65, -1.7]], "bounds_bottom": [[-3.2, 0.0, 1.7], [3.2, 0.0, 1.7], [3.2, 0.0, -0.1], [0.9, 0.0, -0.1],
[0.9, 0.0, -1.7], [-3.2, 0.0, -1.7]], "objects": []</scenegraph>
```

```
Response (Model Output)
'commands': ['<add>dark wooden double bed</add>', '<add>creamy white nightstand</add>', '<add>black floor
lamp</add>', '<add>white two-seat sofa</add>', '<add>low shelf</add>']
```

Figure 16: Example of an input/output pair to the zero-shot LLM on full scene synthesis. Each command gets iteratively processed by ReSpace. For full scene synthesis, this results in an autoregressive loop into SG-LLM such that objects get added into the partial scene.

```
System Prompt
you are a world-class leading interior design expert. your task is to remove furniture given the
descriptions in the header and the current list of furniture in the body. you must respond ONLY with a
valid JSON string that matches precisely the *format* of the existing JSON in the request.
if there are multiple objects that match the description precisely, you should remove all of them.
the prompt for the object to be removed will be given in the header between <remove>...</remove> tags.
the current scene will be given as a JSON object in the body between <scenegraph>...</scenegraph> tags.
in the successful case, your output contains one or N fewer objects in the "objects" list and the rest of
the JSON object should be EXACTLY identical to the input.
you can also remove all objects if the prompt matches those objects. in that case, you provide an empty
list for the "objects" key.
you can further assume that in most cases, there will be at least one object in the scene that matches
the description roughly. this object shall be removed.
only output the JSON (with the removed objects) as a plain string and nothing else.
```

```
User Prompt
<remove>{OBJECT_PROMPT_FOR_REMOVAL}<remove>
<scenegraph>{JSON_DUMP_OF_SSR}</scenegraph>
```

Figure 17: System and User Prompt for the zero-shot LLM for object removal.

```
User Prompt
<remove>comfortable aesthetic bed<remove>
<scenegraph>"room_type": "bedroom", "bounds_top": [[-1.55, 2.6, 1.9], [1.55, 2.6, 1.9], [1.55, 2.6,
-1.9], [-1.55, 2.6, -1.9]], "bounds_bottom": [[-1.55, 0.0, 1.9], [1.55, 0.0, 1.9], [1.55, 0.0, -1.9],
[-1.55, 0.0, -1.9]], "objects": ["desc": "Mid-Century Modern nightstand with light wood finish,
geometric cutout handle, and angled legs.", "size": [0.5, 0.55, 0.43], "pos": [1.37, 0.0, -1.45],
"rot": [0, -0.70711, 0, 0.70711], "jid": "9603344b-99b8-43db-abf0-73c7eaf0ea5f-(0.81)-(1.0)-(0.72)",
"desc": "Modern minimalist TV stand with walnut brown wood, white accents, two closed cabinets, and
open shelving on raised legs.", "size": [1.86, 0.42, 0.35], "pos": [-1.35, 0.0, -0.52], "rot": [0,
0.70711, 0, 0.70711], "jid": "18d54650-68ae-4d4b-8079-1f050b267153-(1.01)-(1.0)-(0.82)", "desc":
"Modern industrial wardrobe with a minimalist metal and mesh design, featuring a rectangular shape,
four slender legs, a single shelf, and a hanging rod.", "size": [1.6, 2.11, 0.5], "pos": [0.74,
0.0, 1.56], "rot": [0, 1, 0, 0], "jid": "19035101-21a1-4495-ae95-90d8d1ccd108-(1.55)-(1.0)-(1.04)",
"desc": "Modern minimalist floor lamp with a white fabric drum shade and brown wooden tripod base
featuring an open geometric design.", "size": [0.47, 1.1, 0.47], "pos": [-1.27, 0.0, 0.6], "rot":
[0, -0.35334, 0, 0.93549], "jid": "4e086af8-0900-4c6d-b535-08c7a936a981-(1.0)-(0.85)-(1.0)",
"desc": "Modern adjustable pendant lamp with branching black metal arms, gold disk shades,
and minimalist design.", "size": [0.36, 1.03, 0.32], "pos": [0.06, 1.79, -0.13], "rot":
[0, 0, 0, 1], "jid": "6289a052-b4a7-417a-b687-b22fddc14663-(0.23)-(1.0)-(0.31)", "desc":
"Mid-Century Modern nightstand with light wood finish, geometric cutout handle, and angled
legs.", "size": [0.5, 0.55, 0.43], "pos": [1.35, 0.0, 0.62], "rot": [0, -0.70711, 0, 0.70711],
"jid": "9603344b-99b8-43db-abf0-73c7eaf0ea5f-(0.81)-(1.0)-(0.72)", "desc": "A contemporary
king-size bed with a brown padded headboard, Hello Kitty-themed pink and white bedding,
graphic pillows, and bolster cushions, offering a playful yet comfortable aesthetic.", "size":
[1.56, 0.9, 2.22], "pos": [0.54, 0.0, -0.43], "rot": [0, -0.70711, 0, 0.70711], "jid":
"bff6e7b4-4e17-401d-b684-dc768def3869-(0.88)-(0.91)-(1.14)"]</scenegraph>
```

```
Response (Model Output)
"room_type": "bedroom", "bounds_top": [[-1.55, 2.6, 1.9], [1.55, 2.6, 1.9], [1.55, 2.6, -1.9], [-1.55,
2.6, -1.9]], "bounds_bottom": [[-1.55, 0.0, 1.9], [1.55, 0.0, 1.9], [1.55, 0.0, -1.9], [-1.55, 0.0,
-1.9]], "objects": ["desc": "Mid-Century Modern nightstand with light wood finish, geometric cutout
handle, and angled legs.", "size": [0.5, 0.55, 0.43], "pos": [1.37, 0.0, -1.45], "rot": [0, -0.70711,
0, 0.70711], "jid": "9603344b-99b8-43db-abf0-73c7eaf0ea5f-(0.81)-(1.0)-(0.72)", "desc": "Modern
minimalist TV stand with walnut brown wood, white accents, two closed cabinets, and open shelving
on raised legs.", "size": [1.86, 0.42, 0.35], "pos": [-1.35, 0.0, -0.52], "rot": [0, 0.70711,
0, 0.70711], "jid": "18d54650-68ae-4d4b-8079-1f050b267153-(1.01)-(1.0)-(0.82)", "desc": "Modern
industrial wardrobe with a minimalist metal and mesh design, featuring a rectangular shape, four
slender legs, a single shelf, and a hanging rod.", "size": [1.6, 2.11, 0.5], "pos": [0.74, 0.0,
1.56], "rot": [0, 1, 0, 0], "jid": "19035101-21a1-4495-ae95-90d8d1ccd108-(1.55)-(1.0)-(1.04)",
"desc": "Modern minimalist floor lamp with a white fabric drum shade and brown wooden tripod base
featuring an open geometric design.", "size": [0.47, 1.1, 0.47], "pos": [-1.27, 0.0, 0.6], "rot":
[0, -0.35334, 0, 0.93549], "jid": "4e086af8-0900-4c6d-b535-08c7a936a981-(1.0)-(0.85)-(1.0)",
"desc": "Modern adjustable pendant lamp with branching black metal arms, gold disk shades,
and minimalist design.", "size": [0.36, 1.03, 0.32], "pos": [0.06, 1.79, -0.13], "rot":
[0, 0, 0, 1], "jid": "6289a052-b4a7-417a-b687-b22fddc14663-(0.23)-(1.0)-(0.31)", "desc":
"Mid-Century Modern nightstand with light wood finish, geometric cutout handle, and angled legs.",
"size": [0.5, 0.55, 0.43], "pos": [1.35, 0.0, 0.62], "rot": [0, -0.70711, 0, 0.70711], "jid":
"9603344b-99b8-43db-abf0-73c7eaf0ea5f-(0.81)-(1.0)-(0.72)"]
```

Figure 18: Example of an input/output pair to the zero-shot LLM on object removal.

