# OpenReview forum: "ReSpace: Text-Driven 3D Indoor Scene Synthesis and Editing with Preference Alignment"
_ICLR.cc/2026/Conference — Submitted to ICLR 2026_

### Official Review · Reviewer_EZ1o · 2025-10-22

**Soundness:** 2
**Presentation:** 2
**Contribution:** 2
**Rating:** 4
**Confidence:** 4

**Summary:**

This paper proposes a next-token prediction pipeline for language-driven indoor scene editing.  Specifically, a preference optimization with reinforcement learning with verifiable rewards is used for finetuning. Additionally, a voxel-based loss function metric is used for capturing geometric interactions beyond bounding boxes. Experiment results show better results on object addition in indoor scene synthesis.

**Strengths:**

This paper has proposed several interesting modules for object addition in given indoor scenes, which includes:
1. a compact structured scene representation with explicit room boundaries that enables asset-agnostic placement
2. voxelization-based Loss rather than bbox constraint

**Weaknesses:**

1. It is not clear how Group Relative Policy Optimization (GRPO) is used for training the AR model (L211-212). There is no motivation of using this and advantage $A_i$ is not explained either, which is claimed to be the contribution of the paper.
2. Computation cost comparison is missing.
3. I do not find where the overall loss fuction is designed,  since there is envoved with several loss functions.
4. How to enable the proposed OOB and MBL loss functions in training/optimization the model? The details are not clearly explained.
5. Missing discussion and comparison with other related autoregressive based indoor scene synthesis/editing works such as:
[1] FOREST2SEQ: Revitalizing Order Prior for Sequential Indoor Scene Synthesis, https://arxiv.org/abs/2407.05388
[2] CASAGPT: Cuboid Arrangement and Scene Assembly for Interior Design, https://arxiv.org/abs/2504.19478

**Questions:**

Some parts are not explained clearly:
1. the modeling of rectilinear polygons in L161
2. What is the differences among A, B and L in L303?
3. How to calculate PMS when ATISS and Mi-Diff do not support language-driven editing?
4. Curious about whether the model support scene rearrangement.
5. Any visual limitations?

---

> ### Author Response · Authors · 2025-11-26
>
> Dear Reviewer EZ1o,
>
> We thank you for your review and recognition of our compact SSR and voxelization-based evaluation. We address your concerns below and have updated the manuscript with additional clarifications.
> \
> \
> **1. On using GRPO for training**
>
> We appreciate this question and want to clarify our positioning. GRPO itself is not our contribution: it was proposed by Shao et al. (2024) and has been popularized in LLM post-training, as we mention in the final paragraph of our Related Work section (L140-148).
>
> Our contribution is being the first to apply preference alignment with verifiable rewards (RLVR) to the scene synthesis task. We use the standard HuggingFace implementation of GRPO and keep the technical description brief to save space, providing full training details in Appendix A.2 (referenced at L278 in Section 4.1).
>
> The advantage of using GRPO in our setting is that it allows us to leverage verifiable rewards like VBL, PMS, and size similarity to align the model toward higher-quality placements beyond what supervised fine-tuning alone achieves.
> \
> \
> **2. Missing computation cost**
>
> This is an excellent point. Following Reviewer fzd3's similar concern, we conducted runtime analysis on 50 full scenes and added Table 6 in Appendix A.7:
>
> | Method | Runtime (s) |
> |--------|-------------|
> | ATISS | 0.54±0.17 |
> | Mi-Diff | 3.79±0.22 |
> | ReSpace (BoN=1) | 8.70±7.21 |
> | LayoutGPT | 10.72±4.33 |
> | ReSpace (BoN=8) | 26.21±25.22 |
> | LayoutVLM | 42.89±16.88 |
>
> **ReSpace is actually faster than LayoutGPT** (8.70s vs. 10.72s), and even with BoN=8 test-time scaling, remains competitive (26.21s vs. LayoutVLM's 42.89s). We note significant room for further speedups via vLLM, quantization, and optimized VBL computation.
> \
> \
> **3 & 4: Loss function and MBL/OOB involvement**
> We understand the confusion here and want to clarify the training setup. For supervised fine-tuning (SFT), we use standard next-token prediction with cross-entropy loss on ground-truth tokens (standard loss for LLMs). For GRPO, we use the objective shown in Equation 2 with verifiable rewards. As outlined in L460-467 and detailed in Appendix §A.2, VBL (which combines OOB + MBL) serves as an evaluation metric and as a binary reward filter during GRPO training, not as a direct loss during training.
>
> Specifically, candidates with VBL < 1e-5 receive reward +1.0, while those failing our quality filters receive reward -1.0. We cannot use VBL as a direct differentiable loss because it involves discrete voxelization—this is why we use it as a verifiable reward signal in the preference alignment stage. The overall loss is the GRPO objective (Equation 2), which optimizes the policy using these rewards.
> \
> \
> **5. Missing discussion with FOREST2SEQ and CASAGPT**
>
> Thank you for pointing out these works. We have now added them to our Related Work section. FOREST2SEQ focuses on ordering strategies for autoregressive synthesis but does not support text-driven editing or explicit boundary handling for complex layouts. CASAGPT targets cuboid arrangement but lacks natural language object descriptions and editing capabilities.
>
> Our work differs by combining autoregressive synthesis with structured text representation, explicit non-rectangular boundaries, and text-driven editing operations (add/remove/swap). To clarify our positioning with recent methods, we added **Table 1** and compare key properties across recent methods, highlighting that ReSpace uniquely combines all five capabilities.
> \
> \
> **6. Additional clarifications**
>
> Regarding your specific questions: The rectilinear polygon extraction (L161) is detailed in Appendix A.1 as Algorithm 1. The notation /A, /B, /L (L303) refers to our training splits: /A = "all", /B = "bedroom", /L = "livingroom", as defined in Section 4.
>
> For PMS calculation, we compute it for baselines on single-object addition by mapping the object prompt to the one-hot class label the baseline expects. For full scene synthesis, baselines don't take prompts as input, making PMS computation not applicable (hence "n/a" in Table 2).
>
> Scene rearrangement is not directly implemented but represents interesting future work—different runs with the same object list naturally yield different spatial arrangements, though instruction-driven rearrangement would require complex data augmentation with user instructions and corresponding ground-truth rearrangements.
>
> Regarding visual limitations, we show diverse failure cases and outputs throughout the paper, with extensive qualitative examples in Figure 13 in the Appendix.
> \
> \
> We hope these clarifications address your concerns and would be happy to provide any additional details you find helpful.

---

### Official Review · Reviewer_pMta · 2025-10-29

**Soundness:** 3
**Presentation:** 3
**Contribution:** 2
**Rating:** 4
**Confidence:** 4

**Summary:**

The paper proposes ReSpace, a text-driven framework for 3D indoor scene synthesis and editing built around a compact Structured Scene Representation (SSR), a specialized SG‑LLM for object addition, and a zero-shot LLM that decomposes user instructions and performs object removal via direct SSR edits. The system adds objects autoregressively, decouples asset selection from layout via a probabilistic sampler, and evaluates layouts with a new Voxelization‑Based Loss (VBL) that counts out‑of‑bounds voxels and mesh‑mesh overlaps.
The SG-LLM is trained in a dual-stage pipeline: first with Supervised Fine-Tuning (SFT) and then with preference alignment (using GRPO), where the VBL metric serves as a verifiable reward signal. Experiments show that ReSpace achieves state-of-the-art results on object addition and, despite mixed results on rendering-based metrics (FID/KID), achieves superior human-perceived quality for full scene synthesis, as validated by a large-scale user study.

**Strengths:**

1. Editable Generative Representation: The SSR (a JSON format) and SG-LLM successfully frame 3D scene generation as an editable, next-token prediction task.
2. Accurate, Rewarded Alignment: The framework leverages GRPO, using the novel VBL metric as a verifiable reward, to fine-tune the model for more geometrically accurate placements.
3. Superior Performance: ReSpace achieves state-of-the-art results on object addition and, more importantly, is rated by human evaluators as having superior perceived quality for full scene synthesis.

**Weaknesses:**

1. Lacks other baselines: The other LLM-based baselines, such as LayoutVLM, which also fine-tune the LLM for scene configuration generation, are not included in the paper. Besides, for the scene synthesis task, at least one end-to-end method needs to be compared in the paper, like InstructScene.
2. Modest Impact of GRPO: The paper highlights the SFT+GRPO pipeline as a key contribution. While Table 1 shows GRPO improves quantitative VBL metrics, the appendix (A.6) reveals a crucial finding: a second user study comparing SFT-only vs. SFT+GRPO found no statistically significant human preference (51% vs 49%). This significantly weakens the claim that preference alignment is a key driver of perceived quality. The authors also note "training fragility" and "reward hacking", suggesting this component is difficult to implement and its benefits may be marginal.
3. Limited Editing Capability: The framework is marketed for "synthesis and editing," but the editing functionality is split and weak. Object removal is handled by the ZS-LLM with low accuracy (75.2% on the ‘liv’ split). More complex edits like "move," "rotate," or "resize" are not supported at all, as the authors admit.

**Questions:**

1. The removal accuracy is a major bottleneck. Is this failure primarily due to semantic ambiguity (e.g., two "chairs" in the scene), or does the ZS-LLM also fail at the task of correctly manipulating the JSON string?
2. For full scene synthesis, why does the model handle the objects one by one? Can we directly generate the SSR for the full scene?
3. What is the per‑object addition latency (SG‑LLM + retrieval + VBL check) and how does BoN scale in interactive settings?

---

> ### Author Response · Authors · 2025-11-26
>
> Dear Reviewer pMta,
>
> Thank you for your review and recognition of our editable SSR representation, GRPO alignment with verifiable rewards, and superior human-perceived quality. We address your concerns below and have updated the manuscript.
> \
> \
> **1. Lacks of other baselines**
>
> Following your suggestion, we implemented LayoutVLM as an additional baseline and added Table 1 comparing key properties across recent methods, highlighting that ReSpace uniquely combines all five capabilities. Updated Table 2:
>
> | Method | FID ↓ | FID_CLIP ↓ | KID_×1e3 ↓ | OOB_×1e3 ↓ | MBL_×1e3 ↓ | VBL_×1e3 ↓ |
> |--------|-------|------------|------------|------------|------------|------------|
> | LayoutGPT | 106.75±0.5 | 9.17±0.1 | 38.97±1.0 | 1199.7±57.6 | **84.21±6.0** | 1284.0±63.3 |
> | LayoutVLM | 80.04±0.6 | 5.91±0.1 | 6.33±0.4 | **78.6±2.2** | 84.34±3.6 | **162.9±5.2** |
> | **ReSpace/A†** | **70.86±0.9** | **4.57±0.2** | **2.07±0.0** | 88.8±7.4 | 124.60±23.6 | 213.4±24.3 |
>
> However, this comparison favors LayoutVLM in three ways: (1) LayoutVLM requires ground-truth objects with bounding boxes as input; ReSpace generates everything from scratch. (2) LayoutVLM is a constraint solver optimized to minimize violations through iterative optimization; ReSpace learns spatial reasoning as a generative model. Despite this, our VBL remains competitive (213 vs. 163). (3) LayoutVLM cannot perform text-driven editing, a core contribution of our work.
>
> ReSpace significantly outperforms LayoutVLM on FID (70.86 vs. 80.04) and KID (2.07 vs. 6.33), indicating superior scene realism and diversity. Our fully generative approach offers key advantages: inherent size generation, scalability with larger datasets, and flexibility for complex non-rectangular layouts and text-driven editing. Our human evaluation (Table 5, A.6) demonstrates 50.3-54.2% win rates versus 38.6-41.9% for ATISS and Mi-Diff, validating our approach.
> \
> \
> **2. Modest impact of GRPO**
>
> We agree the human preference study shows modest differences (51% vs 49%). We do not claim it's a "key driver"—our 54.2% win rate likely comes from overall system design. Instead, we argue "preference alignment can positively influence human-perceived scene quality without degrading spatial reasoning capabilities" (L992-994). We're transparent about these nuanced results (L364-366), showing where GRPO helps quantitatively (VBL) while having modest human preference impact. Reviewer fzd3 noted that "preference alignment via GRPO is novel, and seems to achieve good results," supporting our exploration.
> \
> \
> **3. Limited editing capability and removal accuracy**
>
> We conducted additional analysis (A.8, Figure 14) examining three hypotheses: (1) semantic ambiguity with multiple same-class objects, (2) longer SSR sequences challenging text editing, and (3) evaluation methodology requiring all matching objects removed.
>
> Our key findings: Removal accuracy drops dramatically with SSR length—95% at <200 words to <35% at >500 words, explaining the 'liv' split performance gap. Longer prompts (7 words) achieve 100% accuracy versus ~75% for shorter prompts. Analyzing 326 failures shows "different class" errors (182, 56%) as the primary bottleneck—fundamental reasoning failures rather than ambiguity.
>
> The 'liv' split uses an 8B model due to compute constraints. We hypothesize larger models (>70B) would substantially improve performance on JSON manipulation.
>
> Regarding move, rotate, resize: we agree these represent valuable future work. However, even fundamental add/remove operations seem to pose significant challenges (removal accuracy 75-90%, BoN scaling needed for addition). More advanced editing operations require extensive parametric instruction augmentation, view-dependent handling ("left of X"), and don't align with our real-world deployment goals (L467) (e.g., resize doesn't translate to physical furniture).
> \
> \
> **4. On generating full SSR directly**
>
> We could theoretically train SG-LLM for complete multi-object generation. However, as mentioned in L245-250, this requires more complex fine-tuning to prevent mode collapse from task/class imbalance. We focused on high-quality single object placement where spatial reasoning matters most. We find that our zero-shot LLM provides sufficient baseline for removal and decomposition as proof of concept.
> \
> \
> **5. On per-object addition latency and BoN scaling**
>
> We added a runtime analysis in Table 6 (A.7):
>
> | Method | Runtime (s) |
> |--------|-------------|
> | ATISS | 0.54±0.17 |
> | Mi-Diff | 3.79±0.22 |
> | ReSpace (BoN=1) | 8.70±7.21 |
> | LayoutGPT | 10.72±4.33 |
> | ReSpace (BoN=8) | 26.21±25.22 |
> | LayoutVLM | 42.89±16.88 |
>
> Single object addition for ours is relatively fast, and remains competitive with 1.96s per object on average with BoN=1, and 8.56s for BoN=8. Significant speedups are possible via vLLM, quantization, and optimized VBL.
> \
> \
> We hope these clarifications address your concerns and would be happy to incorporate any further suggestions you have.

---

### Official Review · Reviewer_xQ5n · 2025-10-30

**Soundness:** 2
**Presentation:** 2
**Contribution:** 2
**Rating:** 2
**Confidence:** 3

**Summary:**

This paper proposes a text-driven scene editing method using LLM. The proposed system employs dual-LLM architecture: a primary LLM acts as a natural language interface to translate user instructions into specific prompts, and a second, specially fine-tuned Scene-Graph LLM (SG-LLM) generates and edits a Structured Scene Representation (SSR) based on those prompts. The SG-LLM is trained using supervised fine-tuning followed by preference alignment. The authors demonstrate state-of-the-art performance on the 3D-FRONT dataset.

**Strengths:**

The authors introduces a text-to-indoor scene editing method by leveraing LLMs to autoregressively predict next object with preference alignment.
The paper proposes a new evaluation metric Voxelization-Based Loss (VBL) to measure fine-grained geometric interactions among the room boundaries and 3D objects.

**Weaknesses:**

Lack of Representation Justification: The paper claims to use a Structured Scene Representation (SSR), but it does not sufficiently justify its advantages over other contemporary scene representations formulated as language, such as those in LayoutGPT [1], SceneScript [2], or SpatialLM [3]. Furthermore, encoding scene boundaries is not novel, as methods like Ctrl-Room [4] explicitly model complex walls as part of the generative process. In contrast, this work treats boundaries only as a fixed input. A comparative analysis of SSR's expressiveness and efficiency is needed.

Unmotivated Dual-LLM Architecture: The rationale for the two-LLM pipeline is unclear. A critical question is whether the SG-LLM, trained specifically on SSR data, is prone to overfitting and fails to generalize to direct natural language input, thus necessitating the first LLM as an intermediary. This relates to a core challenge in domain-specific LLMs. The authors should analyze this potential issue and strengthen their experimental validation by including more complex and realistic datasets like Structured3D[5] or the one used in SpatialLM [3] to test the model's robustness.

Limited Experimental Scope: the chosen baselines are not the most recent, and newer methods like InstructScene[6] and SceneWeaver[7] also support intuitive, text-driven editing. More importantly, the 3D-FRONT dataset is known for its simplicity in room diversity and layout complexity. To fully establish the method's efficacy, experiments on more challenging datasets with greater object count and layout variety are encouraged.

Some minor weaknesses:
•Inconsistency in Claims: The authors criticize previous works for being limited by simplified object categories and fixed floor plans. However, the proposed method itself operates under similar constraints, as it starts from a partial scene and retrieves objects from a fixed dataset. The authors should more clearly articulate how their method advances beyond these limitations.
•Clarification of Formulation: Equation 1 requires clarification.
1) What does the p_i represent ?
2) Is \mathcal U_i equivalent to Tok(S_i) ? The relationship between these terms should be explicitly stated.
•Clarification of Prompt Bank: The role and composition of the prompt bank \mathcal P(o) are not well-explained. How does it relate to the sequence modeling process defined in Equation 1?


[1] LayoutGPT: Compositional Visual Planning and Generation with Large Language Models
[2] SceneScript: Reconstructing Scenes With An Autoregressive Structured Language Model
[3] SpatialLM: Training Large Language Models for Structured Indoor Modeling
[4] Ctrl-Room: Controllable Text-to-3D Room Meshes Generation with Layout Constraints
[5] Structured3D: A Large Photo-realistic Dataset for Structured 3D Modeling
[6] InstructScene: Instruction-Driven 3D Indoor Scene Synthesis with Semantic Graph Prior

**Questions:**

See my discussion in weakness

---

> ### Author Response · Authors · 2025-11-26
>
> Dear Reviewer xQ5n,
>
> Thank you for your thorough review. We address your concerns below and have updated the manuscript accordingly.
> \
> \
> **1. Lack of representation justification**
>
> We want to emphasize that we do not claim SSR as a technical contribution of our method but rather a design choice. As stated in L172, we explicitly acknowledge taking inspiration from prior work on DSLs such as SceneScript and adopting a simpler approach for ours. Our contribution lies in leveraging this representation for text-driven editing via autoregressive language modeling with preference alignment. Reviewer fzd3 notes that "sequential synthesis is a natural way to support editing tasks" compared to global optimization, supporting our design choice of autoregressive object placement.
>
> While Ctrl-Room models walls explicitly, it focuses on text-to-3D room mesh generation with mask-guided editing, whereas we target furniture placement with explicit non-rectangular boundary support. We use boundaries as conditioning for learned placement (enabling complex layouts), while Ctrl-Room generates the boundaries themselves. We have added Table 1 comparing key properties across recent methods. Generating boundaries for full scene synthesis represents interesting future work.
> \
> \
> **2. Unmotivated dual-LLM architecture**
>
> As stated in L245-250, we write that "While using a single model for all scene synthesis and editing tasks would be theoretically preferable, this would require more complex fine-tuning in an SFT+RL pipeline to prevent mode collapse due to task/class imbalance."
>
> Our dual-LLM approach is a pragmatic choice: we let SG-LLM training specifically focus on high-quality object placement (the core spatial reasoning challenge), while leveraging the general instruction-following capabilities of existing models for removal and decomposition. This is to demonstrate the viability of specialized placement models as a proof of concept. Given that our results still show room for improvement on placement quality (as evidenced by the effectiveness of BoN scaling), it makes sense to focus on this fundamental capability first before tackling the additional complexity of unified multi-task training at once.
> \
> \
> **3. Limited experimental scope**
>
> 3D-FRONT already presents significant challenges: our filtered dataset contains scenes with up to 50 objects and complex non-rectangular layouts—more challenging than simple rectangular subsets used by many methods. Our BoN scaling experiments (Figure 8) show substantial quality improvements with increased compute, indicating unsaturated performance. Structured3D lacks necessary 3D bounding box annotations. SpatialLM was released shortly before submission. We acknowledge InstructScene and SceneWeaver as valuable related work but note they do not support direct text-driven editing—InstructScene requires manual masking for diffusion; SceneWeaver does not show interactive text-driven editing and does not support non-rectangular rooms. Table 1 clarifies our positioning.
> \
> \
> **4. Inconsistency in claims**
>
> Our criticism (L120-125) states prior work "oversimplifies object semantics through one-hot class encodings" AND/OR "are limited to rectangular layouts"—two separate limitations, not both simultaneously. We thoroughly acknowledge our limitations in L469-473. The distinction is that prior methods either lack rich semantic descriptions or cannot handle complex non-rectangular floor plans, whereas we address both while introducing different trade-offs. Please check Table 1 for a comparison.
> \
> \
> **5. Clarification of formulation (Equation 1)**
>
> We have added clarifications to Section 3.2: $p_{i}$ is the object prompt for the next object. $\text{Tok}(S_{i})$ is the complete token sequence for current scene SSR. $U_{i}$ is the token sequence for the next object only, while $U_{\text{prev}}$ is the existing SSR token sequence (excluding current object). Thus, $\text{Tok}(S_i) = U_{\text{prev}} + U_i$.
> \
> \
> **6. Clarification of prompt bank**
>
> The prompt bank $P(o)$ serves as data augmentation for object prompts, providing varying descriptions with different levels of detail for the same object. For example, a single bed might have prompts ranging from "bed" (1 word) to "modern king-size platform bed" (4 words) to "contemporary king bed with upholstered headboard" (6 words). This diversity ensures the model learns a robust prompt-to-object mapping rather than memorizing specific phrasings. During training, we sample $p_i \sim \text{Unif}(P(o))$ for each object, exposing the model to the full range of prompt styles. This relates to the sequence modeling in Equation 1 as the conditioning signal $p_i$ that guides object generation. We have added this clarification to Section 4.
> \
> \
> We hope these clarifications address your concerns and would be grateful for any suggestions on strengthening the manuscript further.

---

### Official Review · Reviewer_fzd3 · 2025-10-31

**Soundness:** 3
**Presentation:** 3
**Contribution:** 3
**Rating:** 4
**Confidence:** 4

**Summary:**

The paper introduces ReSpace, a framework for autoregressive indoor scene generation and editing from natural language prompts, representing the current scene with a Structured Scene Representation (SSR). The framework is designed to perform three main tasks. For (a) object removal, a zero-shot LLM directly edits the SSR, and for (b) full scene generation, the same zero-shot LLM produces object prompt list, which is fed to SG-LLM. SG-LLM is trained for (c) single object addition with SFT+GRPO, which predicts the next object placement given the SSR and an object prompt. Additionally, a voxelization-based loss is introduced to capture fine-grained geometric details that bounding-box metrics fail to reflect.

**Strengths:**

1) Prior work on scene synthesis predominantly employs global optimization, which is not well-suited to scene editing. Sequential synthesis is a natural way to support editing tasks.
2) The paper introduces a voxelization-based loss that captures fine-scale details and evaluates spatial arrangements more accurately than bounding-box metrics.
3) Preference alignment via GRPO is novel, and seems to achieve good results.

**Weaknesses:**

1) Missing comparison with recent baselines: ATISS (NeurIPS 2021), LayoutGPT (NeurIPS 2023), and Mi-Diff (2024) are relatively old for full scene synthesis evaluation, making it difficult to assess the proposed model’s performance in the current landscape. More recent baselines such as LayoutVLM [2] report stronger results than those selected here.
2) Generation time: Autoregressive generation is likely slower than in-context learning methods like LayoutGPT. However, end-to-end generation time for a full scene is not reported.
3) Lack of global re-optimization/correction: Although an autoregressive approach is natural for editing, as stated in L460–462, it is not directly suitable without a global correction/re-optimization step (e.g., when there is no space for a large object). A global layout solver or feedback mechanism applied before each insertion could mitigate these limitations and achieve the best of both worlds.

[1] Sun, F. Y., Liu, W., Gu, S., Lim, D., Bhat, G., Tombari, F., ... & Wu, J. (2025). Layoutvlm: Differentiable optimization of 3d layout via vision-language models. In Proceedings of the Computer Vision and Pattern Recognition Conference (pp. 29469-29478).

**Questions:**

All of my concerns are listed in the weaknesses section, and I may adjust the rating if they are well addressed.

---

> ### Author Response · Authors · 2025-11-26
>
> Dear Reviewer fzd3,
>
> We thank you for your thoughtful and constructive review. We appreciate your recognition of our sequential synthesis approach, voxelization-based evaluation, and experiments on preference alignment. We address your concerns below and have updated the manuscript accordingly.
> \
> \
> **1. Missing comparison with recent baselines**
>
> We acknowledge the evolving landscape of LLM-based scene generation. Our method specifically targets **editing via addition/removal for complex, non-rectangular floor plans**—a key limitation that motivated this work. To clarify our positioning, we added **Table 1** comparing key properties across recent methods, highlighting that ReSpace uniquely combines all five capabilities.
>
> Per your suggestion, we implemented **LayoutVLM** as an additional baseline for rectangular floor plans. Updated results in **Table 2** (bottom section):
>
> | Method | FID ↓ | FID_CLIP ↓ | KID_×1e3 ↓ | OOB_×1e3 ↓ | MBL_×1e3 ↓ | VBL_×1e3 ↓ |
> |--------|-------|------------|------------|------------|------------|------------|
> | LayoutGPT | 106.75±0.5 | 9.17±0.1 | 38.97±1.0 | 1199.7±57.6 | **84.21±6.0** | 1284.0±63.3 |
> | LayoutVLM | 80.04±0.6 | 5.91±0.1 | 6.33±0.4 | **78.6±2.2** | 84.34±3.6 | **162.9±5.2** |
> | **ReSpace/A†** | **70.86±0.9** | **4.57±0.2** | **2.07±0.0** | 88.8±7.4 | 124.60±23.6 | 213.4±24.3 |
>
> However, this comparison favors LayoutVLM in three critical ways: (1) Input requirements: LayoutVLM requires ground-truth objects with bounding boxes as input. ReSpace generates everything from scratch (room → object list → autoregressive placement), making this not an apples-to-apples comparison. (2) Method: LayoutVLM is a constraint solver explicitly optimized to minimize layout violations through iterative optimization. ReSpace is a fully generative model that learns spatial reasoning from data. Despite this fundamental difference in approach, our VBL remains competitive (213 vs. 163). (3) Capabilities: LayoutVLM cannot perform text-driven editing (add/remove/swap objects), which is a core contribution of our work (as acknowledged in your review).
>
> ReSpace significantly outperforms LayoutVLM on FID (70.86 vs. 80.04) and KID (2.07 vs. 6.33), indicating superior scene realism and diversity. Moreover, our fully generative approach offers key advantages: (1) inherent size generation of objects without explicit selection, (2) scalability: performance can potentially improve once we train on larger datasets, and (3) flexibility: supports complex non-rectangular layouts and text-driven editing that optimization-based methods cannot handle. We believe the comparison validates our design: we remain competitive on layout metrics despite being generative rather than optimization-based, while delivering superior distribution matching and the editing capabilities your review identified as valuable. To clarify our positioning in comparison with recent methods, we have added **Table 1** to the paper (see updated PDF).
> \
> \
> **2. Generation time**
>
> Excellent point. We conducted runtime analysis on 50 full scenes for each method and added it under **Table 6** in the Appendix §A.7:
>
> | Method | Runtime (s) |
> |--------|-------------|
> | ATISS | 0.54±0.17 |
> | Mi-Diff | 3.79±0.22 |
> | ReSpace (BoN=1) | 8.70±7.21 |
> | LayoutGPT | 10.72±4.33 |
> | ReSpace (BoN=8) | 26.21±25.22 |
> | LayoutVLM | 42.89±16.88 |
>
> **ReSpace is actually faster than LayoutGPT** (8.70s vs. 10.72s), and even with BoN=8 test-time scaling, remains competitive (26.21s vs. LayoutVLM's 42.89s). Further speedups are possible via vLLM, quantization, and optimized VBL computation.
> \
> \
> **3. Lack of global re-optimization/correction**
>
> We fully agree this is a promising direction. However, we note an important trade-off for optimization on full layouts: global optimization tends to converge to local optima and reduce diversity (if having the same start configuration), whereas a fully generative approach can sample multiple distribution modes. Thus, we propose two complementary directions, now incorporated in the revised conclusion: (1) Local correction via optimization: After each autoregressive placement, small optimization steps could eliminate OOB/intersections while preserving generative diversity. Unlike global optimization, these lightweight corrections would fix constraint violations without the convergence issues that hurt mode coverage. (2) Tree-based search (MCTS): As mentioned in our (existing) conclusion, MCTS can explore alternative placement choices through "removal" actions and different branches, recovering from poor early decisions that neither pure generation nor local optimization can address. We believe that combining both approaches — local correction for smaller misplacements and MCTS for overall decision-making — could offer the most promising path forward.

---

### Author Response · Authors · 2025-12-01
**Summary of Revisions & Responses by Authors**

Dear ACs & Reviewers,

We sincerely thank all reviewers for their feedback. We substantially expanded the experimental evaluation, added a major new baseline (LayoutVLM), introduced full runtime analysis, and performed failure-mode studies on object removal. These additions directly address all major reviewer concerns regarding baselines, efficiency, and editing reliability. Changes are marked in purple in the updated PDF.
\
\
**Major Experimental Additions**

Following reviewer suggestions, we conducted three additional experiments:

* **LayoutVLM Baseline** (Reviewers fzd3, pMta): We implemented LayoutVLM and added results to Table 2. ReSpace significantly outperforms on FID (70.86 vs 80.04) and KID (2.07 vs 6.33), indicating superior scene realism and diversity. While LayoutVLM achieves lower VBL (162.9 vs 213.4), this comparison favors LayoutVLM in three critical ways: (a) it requires ground-truth objects with bounding boxes as input while ReSpace generates everything from scratch, (b) it's a constraint solver explicitly optimized to minimize violations while ReSpace is fully generative, and (c) it cannot perform text-driven editing, which is a core contribution of our work.
* **Runtime Analysis** (Reviewers fzd3, EZ1o): We added Table 6 (Appendix A.7) comparing generation time across 50 full scenes. ReSpace (BoN=1) is faster than LayoutGPT (8.70s vs 10.72s), and even with BoN=8 test-time scaling remains competitive (26.21s vs LayoutVLM's 42.89s), with variance primarily driven by object count and scene size.
* **Removal Accuracy Analysis** (Reviewer pMta): We conducted detailed analysis (Appendix A.8, Fig. 14) examining removal failures. Key findings: (a) accuracy drops dramatically with SSR length (95% at <200 words to <35% at >500 words), explaining the 'liv' split gap, (b) "different class" errors (56% of failures) represent the primary bottleneck via fundamental reasoning failures rather than ambiguity, (c) the 8B model struggles with long-context JSON manipulation. We conclude that the failure mode appears dominated by long-context structured editing of very large scenes.

**Reviewer-Acknowledged Strengths**

Reviewers acknowledged key strengths including our sequential synthesis design for editing (fzd3), editable SSR representation (pMta, EZ1o), voxelization-based evaluation (fzd3, xQ5n, EZ1o), and preference alignment novelty (fzd3, pMta).

Most importantly, our comprehensive human evaluation with 125 participants performing 25,000 pairwise comparisons demonstrates consistent human preference wins of 10–15% margins over all baselines (50.3–54.2% vs 38.6–41.9%; Table 4). This empirical evidence confirms that our design choices produce scenes human evaluators consistently prefer, validating our approach despite slightly higher FID/KID scores on some metrics.
\
\
**Positioning Clarification**

We added Table 1 — comparing key properties across recent methods — to emphasize ReSpace's unique positioning. To our knowledge, no prior method we are aware of demonstrates all five capabilities jointly: (1) non-rectangular layouts, (2) explicit object semantics, (3) text-driven editing, (4) trained placement, and (5) probabilistic asset sampling. While individual methods support specific capabilities, no other method combines all five.
\
\
**Additional Concerns**

* **Dual-LLM Architecture** (Reviewer xQ5n): As stated in L245-250, this is a pragmatic design choice to focus SG-LLM training on high-quality object placement only (the core spatial reasoning challenge) while leveraging existing models for removal and decomposition. Given our results still show room for improvement on placement quality (evidenced by BoN scaling effectiveness), it makes sense to focus on this fundamental capability before tackling much more complex multi-task training.
* **Experimental Scope** (Reviewer xQ5n): 3D-FRONT presents significant challenges—our filtered dataset contains scenes with up to 50 objects and complex non-rectangular layouts, more demanding than the simple rectangular subsets used by many methods. We acknowledge InstructScene and SceneWeaver as related work, but note they do not support direct text-driven editing (InstructScene requires manual masking for diffusion; SceneWeaver does not demonstrate interactive text-driven editing or support non-rectangular rooms). Table 1 clarifies our positioning relative to these methods.
* **GRPO Impact** (Reviewer pMta): We are transparent about GRPO's modest human preference gains (51% vs 49%, Table 5). We do not claim it's a "key driver" of quality: our 54.2% win rate likely comes from overall system design. However, GRPO demonstrably improves quantitative VBL metrics while maintaining human-perceived quality, which we view as valuable given the training fragility we observed.

We believe these revisions substantially address the major concerns raised and strengthen both the experimental foundation and positioning of our work.

---

### Meta-Review · Area_Chair_Lwmr · 2025-12-30

**Summary:**

The reviewers’ concerns primarily focus on the following aspects:
1. Limited experimental comparisons. Missing comparisons with recent state-of-the-art and autoregressive methods.
2. Generation Time. Despite the supplemented comparison experiments on end-to-end generation time, it still demonstrates high computational cost.
3. Limited Efficacy of GRPO. The benefit of GRPO is not statistically significant in human evaluations and its improvement remains unclear.

**Reviewer Concerns:**

The author addressed issues including model architecture selection, dataset choice, and loss function, and supplemented some comparative experiments. But problems like the high computational cost of end-to-end scene generation, the lack of quantitative comparisons with updated works, and the limited improvement achieved by GRPO remain unresolved.

**Reviewer Scores:**

The reviewers xQ5n and EZ1o may raise their scores after the discussion, since most of their concerns are addressed, except for the computation cost.

---

### Decision · Program_Chairs · 2026-01-26

Reject